# Catalyst: Reveal the Geometry of Pruning by Reshaping Neural Network

## Abstract

Structured pruning aims to reduce the computational cost of neural networks by removing entire filters or channels, but conventional regularization-based approaches suffer from unstable pruning dynamics and magnitude bias. In particular, commonly-used regularizers such as L1 and Group Lasso exhibit trivial global minima and fail to align with the geometry of pruning-invariant configurations, leading to a tradeoff between sparsification and model integrity. We propose Catalyst, a novel regularization framework for structured pruning grounded in extended-space optimization and rigorous landscape geometry. Catalyst introduces auxiliary variables to reshape the loss landscape, admitting a nontrivial global minimizer which aligns to the pruning-invariant set, where pruning decisions are lossless by construction. This formulation enables strong regularization without collapsing the model, and induces robust bifurcation dynamics that separate filters into prune-or-preserve groups with wide decision margins. We provide theoretical analysis of the optimization geometry and bifurcation behavior, and demonstrate empirically that Catalyst achieves stable, magnitude-invariant pruning with superior performance across benchmarks. Our work establishes a principled foundation for structured pruning through geometric regularization and extended-space dynamics.

## 1 Introduction

Structure and sparsity in overparametrized neural networks pose fundamental challenges in understanding the dynamics of deep learning. Structured pruning (He & Xiao, 2023), a model compression technique that removes entire filters or channels, impacts both the structure and sparsity of the model while attempting to preserve performance. Regularization techniques such as L1 and Group Lasso (Yuan & Lin, 2006; Wen et al., 2016) are commonly employed to encourage sparsity by shrinking filter weights toward zero, thereby mitigating the performance degradation caused by pruning. Despite their practical effectiveness, these norm-minimization strategies introduce a strong magnitude bias, favoring the removal of small-norm filters—even when such filters are critical to the model's functionality, because they do not alter the relative scale of filter norms during training.

Moreover, traditional regularizers fail to induce a robust decision boundary between pruned and preserved filters. In such settings, small perturbations in filter norms can lead to unstable or inconsistent pruning decisions. Some methods attempt to address this by encouraging prune-or-preserve bifurcation in proxy variables such as filter norms, gates, or masks (Zhuang et al., 2020; Guo et al., 2021; Ma et al., 2022), leading filters to cluster around distinct "preserve" or "prune" states. However, these heuristics often result in sensitive pruning decisions with narrow margins between the two groups. Filters may be mistakenly pruned when their post-regularization norm lies near the decision boundary, further contributing to instability. Dufort-Labbé et al. (2025)

More critically, pruning attempts that use conventional regularizers on parameter weights suffer from a fundamental limitation in their optimization landscape: the only global minimizer is the trivial zero solution. This creates a regularize–prune tradeoff explained in Ding et al. (2021). If regularization is applied too strongly, the model weights collapse toward zero, breaking the network's functionality. On the other hand, if the applied regularization is too weak, the model retains performance during training but fails to sparsify adequately, making pruning destructive and unstable.

Recent approaches restrict the parameter space to identify filters for pruning early and apply regularization selectively (Jiang et al., 2023; Wang et al., 2019b), thereby avoiding excessive sparsification of essential components and avoiding the degenerate global minima of misaligned regularization. However, such approaches leave the fundamental flaw of making pruning decisions on magnitude bias and the unclear separation of pruning filters unaddressed.

We resolve this issue with Catalyst, a novel regularization scheme that also enables lossless structured pruning. Key idea of Catalyst is to prune in an extended parameter space with auxiliary "catalyst" variables, which reshapes the learning dynamics of pruning. Unlike conventional regularizers used in structural pruning, Catalyst admits a special nontrivial global minimizer: the pruning-invariant set $X^{tgt}$, which geometrically characterizes configurations where pruning does not harm model functionality, hence enabling lossless pruning. Also, Catalyst regularization makes pruning dynamics robust, ensuring the distributions of filters bifurcate naturally into prune group or preserve group. As a result, Catalyst supports theoretically robust lossless pruning, where the performance is retained after pruning; whereas often "lossless" in empirical literature refers to destructive pruning followed by successful post-finetuning. We demonstrate the robustness of Catalyst pruning both theoretically and empirically, showing that theoretically well-constructed regularizer can enhance empirical results such as pruning stability, compression performance, and interpretability.

## 2  RELATED WORKS

### 2.1  STRUCTURED PRUNING

Structured pruning methods remove one or more filters, which is slice of target weight parameter along the axis of output dimension, from the given model. Structured pruning methods usually exploit $L_1$ or Group Lasso (Yuan & Lin, 2006; Wen et al., 2016) regularizers to select filters to prune from diverse target layers such as batch normalization layers (Liu et al., 2017; Zhuang et al., 2020; Kang & Han, 2020), convolution layers (Wang et al., 2020; Wu et al., 2024), external layers (Ding et al., 2021; Guo et al., 2021) or multiple filters from different layers (Fang et al., 2023). Our work defines the algebraic condition to achieve lossless pruning and proposes a novel algebraically principled regularizer for structured pruning methods.

To make pruning decisions more robust, structured pruning methods often accompany additional strategies to encourage bifurcation of the values that serve as proxies for pruning decision. (Zhuang et al., 2020) introduces neuron polarization regularizer to bifurcate the filter norms, and more recent works introduce polarization on gates (Guo et al., 2021) or pruning masks (Mo et al., 2023; Ma et al., 2022). Unlike prior works that introduce explicit strategies to encourage bifurcation, our work enjoys provably robust bifurcation based on the algebraically principled regularizer itself, and demonstrates robust bifurcation behaviors befitting the theoretical analysis.

### 2.2  BYPASSING ALGORITHM AS A STRUCTURE REPARAMETERIZATION

Recently, new type of structural reparameterization was proposed in bypassing (Jung & Lee, 2024) which aim to escape the stationary points driven by SGD, by following 3 stages: 1) extending the model by modifying activation, 2) training with algebraic constraint, called comeback constraint, and 3) contracting back to the original model during the training. The comeback constraint of bypassing (Jung & Lee, 2024) is designed to ensure the lossless contraction.

## 3  GEOMETRY OF STRUCTURED PRUNING AND THE CATALYST

We first present geometric formulation of structured pruning in Section 3.1 by defining the condition for lossless pruning and construct a novel algebraically principled regularizer. Next, in Section 3.2 we explain the proposed algorithm and implementation of Catalyst regularizer which is modified from the bypass algorithm introduced in Jung & Lee (2024). Finally in Section 3.3, we present initialization technique for the proposed method with a theoretical analysis of the robust bifurcation dynamics of the novel regularizer.

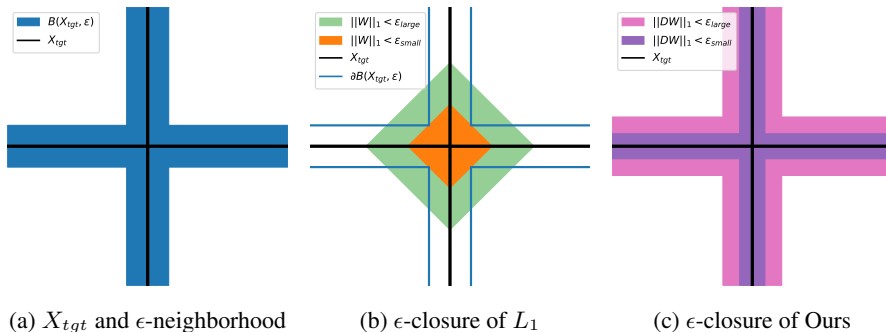

(a) $X_{tgt}$ and $\epsilon$-neighborhood      (b) $\epsilon$-closure of $L_1$      (c) $\epsilon$-closure of Ours

Figure 1: Intuitive plot of geometric mismatch in structured pruning. If $W$ has two channels where individuals are 1-dimensional, the pruning-invariant set $X_{tgt}$ would be union of each axis (black-colored crossing lines).

### 3.1 EQUATION OF LOSSLESS PRUNING

Given weight matrix $W$ and filters $F_i = W_{i,:}$ for $i = 1, \cdots, C$ where $C$ is number of channels, the structured pruning sets $F_i$ to zero for some $i$, to eliminate the $i$th channel.

We formulate the pruning-invariant set $X^{tgt}$, which is a set of weight matrices which is invariant under structured pruning operation, as a finite union of linear subspaces which displayed in Fig. 1a.

$$X_{tgt} := \bigcup_{i=1}^{N_W} \{W | W_{i,:} = F_i = 0\}. \tag{1}$$

If $W$ is in $X_{tgt}$, then $i$th filter $F_i$ is zero for some $i$ and thus we can prune it without any damage.

But, if the regularizer $\mathcal{R}(W)$ is chosen to be L1 or Group Lasso, the $\epsilon$-closure $\{W | \mathcal{R}(W) < \epsilon\}$ would be the ball centered at zero, as visualized in Fig. 1b.

Hence, minimizing $\mathcal{R}(W)$ cannot send $W$ to pruning-invariant set $X_{tgt}$ without collapse of parameters and degradation of model performance due to geometric mismatch. Hence, it inevitably invokes imperfect regularization that the filters cannot be regularized enough before pruning and thus pruning becomes destructive.

To achieve lossless pruning, we need to locate the parameter on $X_{tgt}$ by proper minimization process. Unfortunately, finding direction toward $X_{tgt}$ is difficult. Since $X_{tgt}$ is a union of linear subspaces, the exact defining equations of $X_{tgt}$ would be given by $N_W$ degree multivariate polynomials, which are not able to be efficiently computed or minimized.

This difficulty of minimization is easily resolved by adding *Catalyst*, the additional parameters whose cardinality is $N_W$. With viewpoint of higher dimensional space, we can simplify the defining equation of $X_{tgt}$, by following theorem.

**Theorem 3.1.** *Let $\mathbb{D} = \{D = diag(\delta) | \delta \in \mathbb{R}^{N_W}\}$ be space of diagonal matrices and consider projection map*

$$\begin{aligned} p : \mathbb{R}^{N_W \times N_I} \times \mathbb{D} &\longrightarrow \mathbb{R}^{N_W \times N_I} \\ (W, D) &\longmapsto W \end{aligned} \tag{2}$$

*then*

*(1)  $X_{tgt} = p(\{(W, D) | DW = 0 \text{ and } D \neq 0\})$.*

*(2)  Let $B(X, \epsilon)$ be the $\epsilon$ neighborhood of $X$ with the $L_2$ norm. If $k > 0$ is a positive real number, then*

$$B(X_{tgt}, \epsilon) = p(\{(W, D) | \|DW\|_{2,1} < k\epsilon \text{ and } \|D\|_1 > k\}) \tag{3}$$

In extended space, the defining equation is $DW = 0$, which is just a multiplication of two matrices and can be achieved by minimizing its norm. Note that $D \neq 0$ condition is not necessarily satisfied, since $D = 0$ represents the not-to-prune decision.

In this paper, we propose minimizing

$$\|DW\|_{2,1} = \sum_{i=1}^{N_W} \|(DW)_{i,:}\|_2 = \sum_{i=1}^{N_W} \|D_{ii}F_i\|_2, \tag{4}$$

in extended parameter space as Catalyst regularization. The minimization of $\|DW\|_{2,1}$ has good properties that

1. The global minima of $\|DW\|_{2,1}$ is nontrivial and admits lossless pruning, because if $DW = 0$ and $D \neq 0$ then $W$ must lie in $X_{tgt}$.
2. Although $\|DW\|_{2,1}$ is not convex, all critical points are global minima. Hence, we can reach the global minima by gradient descent.
3. The dynamics of $\|DW\|_{2,1}$ minimization with gradient descent shows wide-margin bifurcation of magnitudes between the pruning filters and the preserving filters. This is described in Section 3.3.

Due to those properties, gradient-descent minimization of $\|DW\|_{2,1}$ ensures the parameters to arrive at $X_{tgt}$ and make pruning lossless: we can pursue to minimize $\|DW\|_{2,1}$ until it arrives to zero, while retaining the model performance. When $\|DW\|_{2,1}$ becomes near-zero, the pruning operation would become harmless. We show this phenomenon empirically in Section 4.5 by showing the performance damage driven by pruning.

Using the catalyst $D$ and $\|DW\|_{2,1}$ requires a proper training algorithm that includes model extension and contraction methods. In Section 3.2 we present Catalyst Pruning algorithm. Interestingly, the constraint for lossless pruning $DW = 0$ is also a constraint for the contraction to original parameter space and allows to remove $D$ at the same time. This is why we call it **Catalyst**.

## 3.2 BYPASSING ALGORITHM FOR CATALYST PRUNING

The algebraic constraint $DW = 0$ for the pruning is defined on extended parameter space with diagonal matrix $D$ in Section 3.1. To exploit this constraint with external parameter $D$, we modify bypass algorithm(Jung & Lee, 2024) to manage the model extension and regularization for the implementation.

Referring (Jung & Lee, 2024), the Bypass pipeline is comprised of the following three core components:

1. $\varphi_2$: Extended model with additional parameter $D$.
2. $embed$: mapping from original model to extended model, by initializing $D$ with function preserving property.
3. $proj$: contraction map which removes $D$ from extended model to restore the original structure, which is function-preserving under specific constraint.

We adopt the Bypass pipeline as the implementation framework for our Catalyst pruning method, with a minor modification to the learnable activation function. Instead of the original form $\psi_D(x) = Dx + \sigma(x)$, we introduce a modified version with two learnable parameters $D$ and $\overline{D}$:

$$\psi_{D,\overline{D}} = Dx - \overline{D}x + \sigma(x) \tag{5}$$

where $\sigma(x)$ denotes the nonlinear activation.

Replacing $\sigma$ to $\psi_{D,\overline{D}}$, we get extended model $\varphi_2$ and embed map $embed : \theta \mapsto (\theta, D^{init}, D^{init})$ which identically initializes $D$ and $\overline{D}$ for function-preserving property. The role of $D^{init}$ would be explained in Section 3.3.

For the $proj$, we propose to use composition of pruning operation and original $proj$ of (Jung & Lee, 2024) which removes $D$ without affecting the neural network under algebraic constraint $ADW = 0$, where $A$ is weight matrix of next layer. We denote this composition by $prune$.

Since $DW = 0$ implies $ADW = 0$, $prune$ becomes function-preserving under $DW = 0$. This allows us to simultaneously prune the regularized filters in $W$ and eliminate the auxiliary parameter

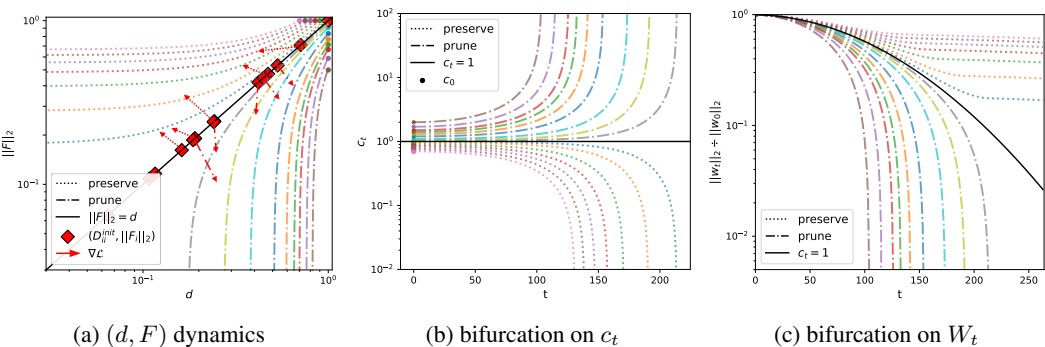

(a) $(d, F)$ dynamics     (b) bifurcation on $c_t$     (c) bifurcation on $W_t$

Figure 2: Simulation on $\|DW\|_{2,1}$ minimization,

$D$ after pruning. As a result, $prune$ will eliminate both $D$ and several filters to be pruned at a same time. The full formulation of $prune$, which is degree 3 polynomial map, is provided in Section A.3 with proof of its function-preserving property.

For the structured pruning, we consider the model compression process consist of $\|DW\|_{2,1}$ minimization and $prune$, which pushes the parameter to satisfy $DW = 0$ and removes the parameters with function-preserving property. Since two parameters $D$ and $\overline{D}$ were exploited, we iterate this process twice to remove $D$ and $\overline{D}$ sequentially, named $opt_1$ and $opt_2$, to recover the original model architecture. The overall algorithm is summarized in Section 4.1.

### 3.3 BIFURCATION BEHAVIOR OF $\|DW\|_{2,1}$ MINIMIZATION

In previous section, we proposed how to prune the model while minimizing $\|DW\|_{2,1}$. To complete the pipeline, we need to choose $D^{init}$ for $embed$. The choice of $D^{init}$ is crucial to control the sparsification, since the destination of $\|DW\|_{2,1}$ minimization is strongly dependent on the initial point $(D^{init}, W)$.

In particular, for each filter $\{F_i\}_{i \in [N_W]}$ of $W$ we propose to set $D_{ii}^{init}$ to $c_0 \|F_i\|_2$ with the hyper-parameter $c_0 = 1$ where $N$ is the number of parameters in $F_i$. For all $i$, if $D_{ii}^{init}$ is set to $\|F_i\|_2$, then $(D_{ii}^{init}, F_i)$ is located on the decision boundary (black line and red diamonds in Fig. 2a) on $(d, F)$-space.

During training, due to performance loss $\mathcal{L}$ (red arrows in Fig. 2a), the $(D_{ii}, F_i)$ escapes this decision boundary almost surely, and then move according to its position whether the ratio $c^{(i)} = \frac{D_{ii}}{\|F_i\|_2}$ is larger than 1 or not. Precisely, according to Theorem 3.2 below, if $c^{(i)} > 1$ then $i$th filter and Catalyst parameter $(D_{ii}, F_i)$ would move toward line $F = 0$ of $(d, F)$-space and thus $i$th filter would be pruned. Otherwise if $c^{(i)} < 1$, then the destination would be the hyperplane $d = 0$ and the $i$th filter would be preserved while $D_{ii}$ becomes zero.

**Theorem 3.2.** *Let $d$ be the scalar parameter and $M$ be N-dimensional vector. $M_t$ and $d_t$ be the trajectory of gradient descent movement of $\|dM\|_2$ at timestep t positive learning rate $\lambda_t$ and weight decay term $\alpha \ll 1$. Let $c_t = \frac{|d_t|}{\|M_t\|_2}$ and assume that*

$$0 < \lambda_* < \lambda_t \ll \min\left(\frac{(1-\alpha)}{c_0}, (1-\alpha)c_0\right) \tag{6}$$

*where $\lambda_* = \inf_t \lambda_t$ is the infimum $\lambda_t$.*

*(1) If $c_0 = 1$, then $c_t = 1$ for all timestep t.*

*(2) If $c_0 < 1$, then $c_t$ exponentially shrinks to $\frac{\lambda_t}{1-\alpha}$. i.e, If T be the smallest integer satisfying $c_T \le \frac{\lambda_T}{1-\alpha}$ then there exists $k \in (0,1)$ such that $c_{t+1} \le kc_t$ for all $t < T-1$.*

*(3) If $c_0 > 1$, then $c_t$ exponentially grows to $\frac{(1-\alpha)}{\lambda_t}$. i.e, if T is the smallest integer satisfying $c_T \ge \frac{1-\alpha}{\lambda_T}$ then there exists $k > 1$ such that $c_{t+1} \ge kc_t$ for all $t < T-1$*

*Proof of Theorem 3.2.* While the sign of each entries of $M$ or d is unchanged, we can find recurrence relation between $c_t$ and $c_{t+1}$. We can prove that multiplication coefficient of recurrence relation depends on the initial value $c_0$ completely, and thus $c_t$ exponentially grows or shrinks. The detailed proofs can be found in Section D. □

The bifurcation behavior explained by Theorem 3.2 is empirically observed in the example dynamics plot in Fig. 2, that bifurcation on $c_t$ in Fig. 2b inherently results in the bifurcation between two classes of filters which would be pushed toward zero (pruned) or preserved, as depicted in Fig. 2c.

This gives rise to a very effective and robust decision strategy to decide which filters would be pruned. As ratio $c_t = \frac{D_{ii}}{\|F_i\|_2}$ bifurcates to getting away from 1, we may prune the $i$th filter when $\frac{D_{ii}}{\|F_i\|_2} > 1$ and construct a set of pruning indices $P = \{i | D_{ii} > \|F_i\|_2\}$.

Since the pruning decision of $i$th filter is made by the position of $(D_{ii}, F_i)$ induced by performance loss $\mathcal{L}$, all channels get provably equal chance to be pruned. That is, filters with smaller initial magnitudes are not any more preferred to be pruned than those with larger magnitudes. We show the real observation of bifurcation of filters in Section 4.3.

# 4 IMPLEMENTATION AND EMPIRICAL VALIDATION

In experiments, we implement Catalyst pruning and verify the expected properties: the lossless pruning and bifurcation behavior. Also, we compare the post-finetuning performance to existing baselines, on image classification benchmark.

## 4.1 PRUNING ALGORITHM: CATALYST PRUNING

We present our implementation of Catalyst algorithm in Algorithm 1 which is adaptation of Bypass pipeline (Jung & Lee, 2024) with catalyst regularization and other theoretical aspects described in Section 3. The algorithm can be summarized as: extend the model $\varphi_1$ by introducing additional parameters $D$ and $\overline{D}$, train in extended space with constrained optimization, and contract back to the original model with pruning.

In detail, the proposed algorithm starts with $embed(D^{init})$, where $D^{init}$ is set to be $D^{init} = c \times diag(\|F_1\|_2, \cdots, \|F_{N_W}\|_2)$ with hyperparameter $c$ with default value 1, as proposed in Section 3.3. The $c = 1$ is proposed to place the pair of $(D_{ii}, F_i)$ on the pruning decision boundary, but the practitioners may set this value to $c > 1$ to prune more, or $c < 1$ to prune less.

After initialization, the proposed algorithm repeats regularize-and-prune loop twice, to remove $D$ and $\overline{D}$ with pruning operation, respectively. During the first loop, namely $opt_1$, we minimize $\mathcal{L} + \gamma_t(\|DW\|_{2,1})$ with SGD optimizer until the training budget $T$.

The $Optimizer_\lambda(\cdot)$ represents the single SGD update with hyperparameter $\lambda$ and $\gamma_t$ is the parameter which controls the weight of regularization as in (Jung & Lee, 2024). The $\gamma_t$ is the coefficient of Catalyst regularizer, which may change according to timestep $t$. We set $\gamma_t$ to increase linearly along $t$.

We can control the pressure of sparsification during $opt_1$, by changing $\alpha_\theta$ and $\alpha_D$ which are the weight decay terms of $\theta$ and $D$ each. Those are considered to be same in Section 3.3, but we may set $\alpha_\theta > \alpha_D$ to promote larger pruning ratio. In case of $\alpha_\theta > \alpha_D$, larger $\gamma_t$ would induce larger pressure on sparsification since the influence of $\nabla \mathcal{L}_2$ is weakened, compared to the deterministic movement of $\nabla \|DW\|_{2,1}$.

If $\|DW\|_{2,1}$ decreases to small positive value $\epsilon$ (line 6 of Algorithm 1) or all $c_t$ are bifurcated enough to satisfy $|log(c_t)| > \kappa$, the regularization loop of $opt_1$ may be stopped early. For experiments, we used $\kappa = 1$ for Imagenet and $\kappa = \infty$ for CIFAR, since early stopping was not necessary. After regularization stage, we choose the pruning indices by threshold $c = 1$, which is equivalent to $P = \{i | D_{ii} > \|F_i\|_2\}$, and prune the selected filters by $prune$ defined in Section 3.2 to obtain intermediate pruning results with one extra parameter $D$ but pruned. Applying similar loop in line 9-

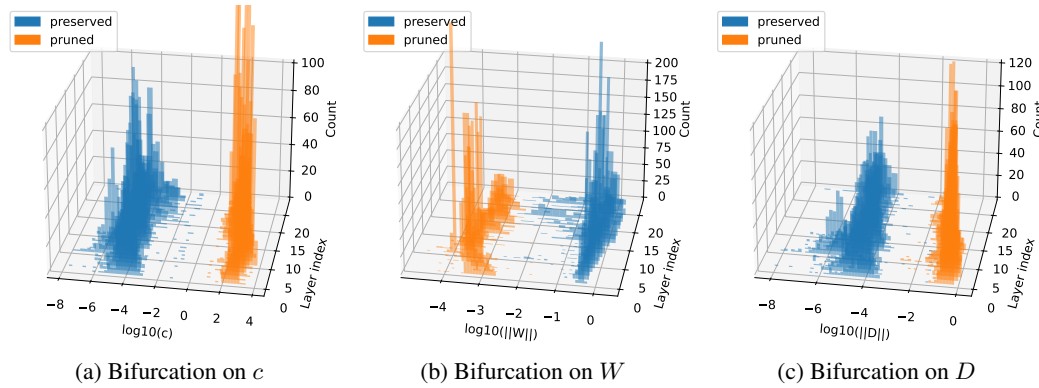

(a) Bifurcation on $c$      (b) Bifurcation on $W$      (c) Bifurcation on $D$

Figure 3: The histograms of ratio $c = \frac{D_{ii}}{\|F_i\|}$, filter vector $F_i$ of weight tensor $W$, and $D_{ii}$'s for each layers of DeiT-T_0.6G model trained on Imagenet. The z-axis represents the frequency.

14 of Algorithm 1, we can prune the model again and obtain pruned model with original architecture.

---

**Algorithm 1:** Regularization with catalyst

---

1: **Input** $\overline{\theta} = (W, b_W, A, b_A), \lambda, \lambda', \mathcal{L}_2, \epsilon, \epsilon', \gamma_t, \gamma'_t, T, T', c = 1$
2: **Initialize** $\overline{\theta}, D, \overline{D} \leftarrow embed(D^{init})(\overline{\theta}, 0, 0)$ where $D^{init} = c \cdot diag(\|F_1\|_2, \cdots, \|F_{N_W}\|_2)$
3: **repeat**
4:     $\overline{\theta}, D, \overline{D} \leftarrow \text{Optimizer}_\lambda(\mathcal{L}_2(\overline{\theta}, D, \overline{D}) + \gamma_t\|DW\|_{2,1})$
5:     $t \leftarrow t + 1$
6: **until** $\|DW\|_{2,1} < \epsilon$ or $|log(c_t)| > \kappa$ or $t > T$
7: $P \leftarrow \{i | D_{ii} > \|F_i\|_2\}$
8: $\overline{\theta}, D, 0 \leftarrow prune(P)(\overline{\theta}, D, \overline{D})$
9: **repeat**
10:     $\overline{\theta}, D, 0 \leftarrow \text{Optimizer}_{\lambda'}(\mathcal{L}_2(\overline{\theta}, D, 0) + \gamma'_t\|DW\|_{2,1})$
11:     $t \leftarrow t + 1$
12: **until** $\|DW\|_{2,1} < \epsilon'$ or $|log(c_t)| > \kappa'$ or $t > T'$
13: $P \leftarrow \{i | D_{ii} > \|F_i\|_2\}$
14: $\overline{\theta}, 0, 0 \leftarrow prune(P)(\overline{\theta}, D, 0)$
15: **return** the pruned parameter $\overline{\theta}$ (and continue finetune.)

---

## 4.2 MODELS, DATASETS AND SETTINGS

We evaluate the Catalyst pruning on image classification benchmark on CIFAR datasets with Resnet56, VGG19bn, and Imagenet with Resnet50 and DeiT-tiny models.

The detailed experimental settings including augmentation and hyperparameters such as learning rate, number of epochs, $\epsilon$ and $T$ can be found in Section E.

Catalyst pruning is applicable to various type of layers such as convolution, fully connected layers, or even grouped multiple layers, as Theorem 3.2 does not limit the dimension of the regularization target dimension. However, we set the scaling factor of BN as a pruning target for CIFAR as first step approach, and concatenate the grouped filters which should be pruned together, for DeiT-tiny and Resnet50-1.96x.

## 4.3 ROBUST BIFURCATION

We present a typically observed bifurcation behavior catalyst regularization in Fig. 3. Fig. 3a shows robust bifurcation of $c = \frac{D_{ii}}{\|F_i\|}$ centered at $c = 1$, by $\|DW\|_{2,1}$ minimization, as theoretically expected in Theorem 3.2. The gap of $c$ between pruned filters and preserved filters is extremely large, around $10^8$ times, since the $c$ value changes exponentially during the minimization.

Table 1: Effect of $prune$ operation in Catalyst pruning and post-pruning accuracy

| Task | avg $\Delta$ acc (%p) | avg $\Delta\mathcal{L}$ (%) | MAC drop (%) | pretrained acc (%) | Post-pruning acc (%) | Post-finetuning acc (%) |
|---|---|---|---|---|---|---|
| R56_2.16x CIFAR10 | 0.001 | 0.0018 | 53.73 | 93.53 | 93.15 | 93.91 |
| V19_8.96x CIFAR100 | -0.005 | 0.0011 | 88.83 | 73.50 | 69.43 | 70.39 |
| V19_11.84x CIFAR100 | -0.021 | 0.0014 | 91.56 | 73.50 | 67.22 | 68.13 |
| V19_13.83x CIFAR100 | 0.004 | 0.0011 | 92.77 | 73.50 | 66.14 | 66.44 |
| R50_2.18x Imagenet | -0.019 | 0.0010 | 54.17 | 76.13 | 74.63 | 75.90 |
| DeiT-T_2.26x Imagenet | -0.257 | 0.0059 | 55.83 | 72.13 | 66.71 | 71.41 |

The bifurcation on $c$ naturally induces the bifurcation on $W$, the filters on the target layer, due to its definition. Evidence of bifurcation on the $W$ bifurcation is visualized in Fig. 3b that the filter norms of pruned filters are pushed to have small values, while the filter norms the rest are preserved. This property is not observed in conventional regularization methods since they push all filters to zero. The bifurcation on $D$ was also observed, that the $D$ of preserved filters sacrifice instead of $W$, and the $D$ value of pruned filters are preserved to be large. The bifurcation behavior was observed on every models. We include all histogram plots of other models in Section H.

### 4.4 MAGNITUDE INDEPENDENT PRUNING

The Catalyst pruning selects the pruning filters according to $c$ after minimizing $\|DW\|_{2,1}$, whether it is larger than 1 or not. This decision is not dependent to the initial magnitude of each filter $F_i$, that small magnitude filter can be chosen to be pruned if corresponding Catalyst variable $D_i$ is reduced faster through training dynamics.

We visualize the distribution of pruned filter's initial magnitude compared to the Lasso and Group Lasso regularization on every layer of VGG19 model with speedup 8.96x, in Section I due to page limit. Unlike magnitude-minimizing regularizers, there is no preference of small magnitude filters in pruning decision, allowing to remove large-magnitude redundant filters from the model.

### 4.5 EMPIRICAL EVIDENCE OF LOSSLESS PRUNING

Due to geometric alignment of $X_{tgt}$ and minimizer of $\|DW\|$, we could apply strong regularization while preserving model performance and make pruning operation becomes harmless. To show this, we report the average of test accuracy drop and test loss difference for each pruning operations Table 1 with total MACs drop.

Pruning at $\|DW\|_{2,1} < \epsilon$ shows promising results of lossless pruning, that the performance drop caused by $prune$ is negligible.

During the regularization, the model performance was retained, as shown by post-pruning performances in fifth column of Table 1 being not severely damaged. Unlike other regularization methods which often shows remarkable performance drop (e.g. Fig. 4 of Ding et al. (2021)) even after the regularization, the Catalyst pruning shows very mild temporary loss during the entire pruning procedure, as shown in Fig. 4 and Fig. 5 in Section F. To the best of our knowledge, pruning methods with such stable loss trajectory has not been reported in the structured pruning literature.

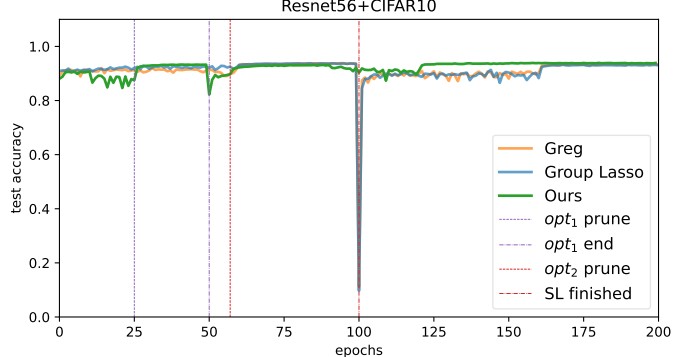

Figure 4: Accuracy graph of R56+CIFAR10 training, comparing Catalyst to Lasso and Greg (Wang et al., 2020).

Table 2: Performance Comparison of Various Filter Pruning Methods after Fine-tuning

| | Method | Baseline ACC(%) | Pruned ACC(%) | Δ ACC(%) | speedup |
|---|---|---|---|---|---|
| | Slimming(Liu et al., 2017; Zhuang et al., 2020) | 93.80 | 93.27 | -0.53 | 1.92x |
| | Polar(Zhuang et al., 2020) | 93.80 | 93.83 | +0.03 | 1.89x |
| | SCP (Kang & Han, 2020) | 93.69 | 93.23 | -0.46 | 2.06x |
| | Hrank(Lin et al., 2020) | 93.26 | 93.17 | -0.09 | 2.00x |
| Resnet56+ CIFAR10 | HBFP (Basha et al., 2024) | 93.26 | 92.42 | -0.84 | 1.77x |
| | ResRep(Ding et al., 2021) | 93.71 | 93.71 | 0.00 | 2.12x |
| | DepGraph(Fang et al., 2023) | 93.53 | 93.77 | +0.24 | 2.13x |
| | SPSRC(Sun & Shi, 2024) | 93.59 | 93.19 | -0.40 | 1.76x |
| | Catalyst-BN | 93.53 | 93.91 | +0.38 | 2.16x |
| | SCP(Kang & Han, 2020) | 72.56 | 72.15 | -0.41 | 2.63x |
| | Greg-1 (Wang et al., 2020) | 74.02 | 71.30 | -2.72 | 2.96x |
| | DepGraph(Fang et al., 2023) | 73.50 | 72.46 | -1.04 | 3.00x |
| | Catalyst-BN | 73.50 | 73.37 | -0.13 | 3.00x |
| | DepGraph(Fang et al., 2023) | 73.50 | 70.39 | -3.11 | 8.92x |
| VGG19+ CIFAR100 | Greg-2 (Wang et al., 2020) | 74.02 | 67.75 | -6.27 | 8.84x |
| | EigenD(Wang et al., 2019a) | 73.34 | 65.18 | -8.16 | 8.80x |
| | Catalyst-BN | 73.50 | 70.39 | -3.11 | 8.96x |
| | DepGraph(Fang et al., 2023) | 73.50 | 66.20 | -7.30 | 12.00x |
| | Catalyst-BN | 73.50 | 68.13 | -5.37 | 11.84x |
| | Catalyst-BN | 73.50 | 66.44 | -7.06 | 13.83x |
| | PFP(Liebenwein et al., 2020) | 76.13 | 75.21 | -0.91 | 1.49x |
| | Greg-1(Wang et al., 2020) | 76.13 | 76.27 | +0.14 | 1.49x |
| | Catalyst-BN | 76.15 | 76.40 | +0.36 | 1.49x |
| | Hrank(Lin et al., 2020) | 76.15 | 74.98 | -1.17 | 1.78x |
| | Slimming(Liu et al., 2017; Zhuang et al., 2020) | 76.15 | 74.88 | -1.27 | 2.13x |
| Resnet50+Imagenet | Polar(Zhuang et al., 2020) | 76.15 | 75.63 | -0.52 | 2.17x |
| | DepGraph(Fang et al., 2023) | 76.15 | 75.83 | -0.32 | 2.08x |
| | OICSR(Li et al., 2019) | 76.31 | 75.95 | -0.37 | 2.00x |
| | Torque(Gupta et al., 2024) | 76.07 | 75.07 | -1.00 | 2.05x |
| | RGP(Chen et al., 2023) | 76.22 | 74.58 | -1.64 | 2.13x |
| | Catalyst-BN | 76.15 | 75.90 | -0.25 | 2.18x |
| | SSViTE (Chen et al., 2021) | 72.20 | 70.12 | -2.08 | 1.44x |
| | WDPruning (Yu et al., 2022) | 72.20 | 70.34 | -1.86 | 1.86x |
| DeiT-T+Imagenet | X-Pruner (Yu & Xiang, 2023) | 72.20 | 71.10 | -1.10 | 2.16x |
| | SNP (Shim et al., 2024) | 72.20 | 70.29 | -1.91 | 2.23x |
| | Catalyst | 72.13* | 71.41 | -0.72 | 2.26x |

## 4.6 PRUNING PERFORMANCE COMPARISON

We present the pruning performance of Catalyst, after substantial finetuning process in Table 2.

We tune the hyperparameters for regularization process to have similar speedup ratio, which is $\frac{\text{(MACs of original model)}}{\text{(MACs of pruned model)}}$, and later applied finetuning. The results with similar pruning ratio or similar speedups are partitioned with horizontal rules, and it is evident that our proposed method achieves promising performance in most partitions.

While our purpose is not just to achieve high post-finetuning performance but gain practical and explainable advantages shown in previous evaluations, the post-finetuning results remain superior to baselines. In detail, Catalyst achieves remarkable improvements in extremely pruned VGG19 and DeiT-Tiny experiments. In VGG19, although we used BN, the naive regularization target with $N = 1$, Catalyst outperformed the modern method DepGraph, which considers the concatenation of multiple connected layers. For DeiT-Tiny, although Catalyst can prune other prunable targets such as embed dimension or attention heads, we aim only one granularity of attention, the neuron level pruning (Shim et al., 2024) which comprises the largest number of channels. As a result, Catalyst could derive superior performance on DeiT-Tiny benchmark.

## 5 DISCUSSION

To present a compression method rather than regeneration, we argue that intermediate performance of regularization-based pruning method should be preserved properly throughout the process, rather than relying on recovery after substantial degradation. The Catalyst pruning shows lossless pruning

---

*Although we used official pretrained model through timm library(Wightman, 2019), the test accuracy was measured to be smaller in our environment. However, it is not considered to be critical since pruned accuracy gap is much larger.

after regularization, and thus provides the smooth learning curve, but however the training-caused damage is observed in Section 4.5. This training-caused damage is inevitable in constrained optimization yet can be readily ameliorated by taking smaller $\gamma_t$ with more training budget, similarly to $opt_2$ of (Jung & Lee, 2024). Alternatively, we may bring other constrained optimization algorithm for the improvement.

Unlike conventional Lasso regularizers, the pruning decision of the Catalyst pruning is independent from the magnitude of the pretrained model. The decision of whether a given filter is pruned depends on whether the ratio $c = \frac{D_{ii}}{\|F_i\|_2}$ falls below or above $1$ during the training process, which follows the gradient of performance loss $\mathcal{L}$. The ratio $c$ has geometric meaning in the extended space, when we define projective space on it, that it is in fact a cotangent of angular distance from embedding of zero filter. As a future work, we will investigate this intuition in projective geometry for another application, with filtered scope of projective space which disregards the role of scale.

## 6 CONCLUSION

We propose a geometric formulation of optimal regularization for lossless structured pruning, and use it as a blueprint to design a novel regularizer with provable potential to achieve lossless pruning. We name the regularizer Catalyst, as it acts like a catalyst for pruning such that the to-be-pruned neural network is temporarily deformed prior to actual pruning operations. We use Catalyst to construct a novel structured pruning algorithm, named Catalyst pruning, which enjoys provable zero-bias fair-chance pruning behavior and robust pruning decision boundary with wide-margin bifurcation dynamics due to theoretical guarantees naturally arising from the Catalyst regularizer. Empirical validations support the theoretically expected benefits of Catalyst regularizer, as Catalyst pruning shows robust bifurcation and magnitude-independent pruning decision while retaining model performance due to theoretically supported lossless pruning property.

## LLM USAGE

We used LLMs for limited purpose for editing the manuscript only. LLMs were not used for any other purposes.

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

# A COMPONENTS OF SECTION 3.2

In this section, we provide detailed explanations on each components of Bypassing, focusing on the modifications for Catalyst pruning.

## A.1 MODEL EXTENSION

Unlike the original bypass algorithm (Jung & Lee, 2024), we propose to use learnable activation which employs two parameters $D$ and $\overline{D}$ for extended model $\varphi_2$, to give nontrivial initialization on $D$, as follows:

**Definition A.1.** *Let $\overline{\theta} = (W, b_W, A, b_A)$ and rewrite*

$$\overline{\varphi_1}(\overline{\theta}) = \overline{\varphi_1}(W, b_W, A, b_A) = NN(W, b_W, A, b_A, \sigma). \tag{A1}$$

1. *We first define learnable activation $\psi_{D,\overline{D}}$ with additional parameter $D$ and $\overline{D}$:*

$$\psi_{D,\overline{D}} : x \mapsto Dx - \overline{D}x + \sigma(x) \tag{A2}$$

2. *We define $\overline{\varphi_2}$ by*

$$\begin{aligned}\overline{\varphi_2}(\overline{\theta}, D, \overline{D}) &= \overline{\varphi_2}(W, b_W, A, b_A, D, \overline{D}) \\ &= NN(W, b_W, A, b_A, \psi_{D,\overline{D}})\end{aligned} \tag{A3}$$

3. *The $\varphi_2(\theta, D, \overline{D})$ is defined by replacing $\overline{\varphi_1}(\overline{\theta})$ to $\overline{\varphi_2}(\overline{\theta}, D, \overline{D})$ from original model $\varphi_1(\theta)$.*

## A.2 THE *embed* FUNCTION

Now we set the *embed* map as follows:

**Definition A.2.** *Given $D^{init}$, we define $embed(D^{init})$ as function-preserving operator on extended parameter space, as follows:*

$$embed(D^{init}) : \overline{\theta} \mapsto (\overline{\theta}, D^{init}, D^{init}) \tag{A4}$$

**Remark A.3.** *The map $embed$ defined in Definition A.2 has a function-preserving property on $\varphi_2$, that $\varphi_2 = \varphi_2 \circ embed(D^{init})$ because $\psi_{D^{init}, D^{init}} = \sigma$.*

For the implementation, we need to choose $D^{init}$ to complete the model extension. In this paper, we proposed to use $D^{init} = diag(\|F_1\|_2, \cdots, \|F_{N_W}\|_2)$, where $N$ is the number of entries in each filter. With the proposed initialization, the fair pruning chance would be ensured and the $\|DW\|_{2,1}$ minimization would show bifurcation behavior. The detailed explanation and mathematical backgrounds which supports this proposed initialization are included in Section 3.3 with simulations.

## A.3 *prune*: COMBINATION OF PRUNING AND *proj*

Now we propose the last core component of Catalyst pruning by defining contraction map *prune*, which replaces *proj* of bypass pipeline.

**Definition A.4.** *Let $P$ be set of filter indices to be pruned and $P^c$ be its complement.*

*We define $prune$ by*

$$\begin{aligned}prune(P) : (W, b_W, A, b_A, D, \overline{D}) &\mapsto \\ (W[P^c], b_W[P^c], (A^T[P^c])^T, b'_A, -\overline{D}[P^c], 0)\end{aligned} \tag{A5}$$

*where $b'_A = b_A + ADb_W + (A^T[P])^T(\psi_{-\overline{D},0}(b_W))[P]$*

**Theorem A.5.** *Let $prune$ be the mappings defined in Definition A.4. If $DW = 0$ and $P = \{i|D_{ii} \neq 0\} = \{i|\forall jW_{ij} = 0\} \in 2^{N_W}$, then $prune(P)$ becomes function-preserving. That is,*

$$\overline{\varphi_2} = \overline{\varphi_2} \circ prune(P). \tag{A6}$$

*Proof of Theorem A.5.* For arbitrary input, if $DW = 0$ then some filters would provide constant features. We pass those constant filters to next layer and add them to bias vector of next layer. The detailed proof is presented in Section C. □

Combining $\varphi_2$, $embed(D^{init})$ and $prune(P)$ we can build the training pipeline for the structured pruning, with constrained optimization algorithm, $opt_1$ and $opt_2$. Starting with $\varphi_1(\theta)$, we extend the model to $\varphi_2(\theta, D, \overline{D})$ by $embed(D^{init})$, train with constrained optimization $opt_1$ and algebraic constraints $DW = 0$, and prune the model by $prune(P)$ to get pruned but still extended model $\varphi_2(\theta[P], D[P], 0)$. Again, with constrained optimization $opt_2$ and $\|DW\|_{2,1} = 0$, we get final pruned model by second $prune$. As stated in Remark A.3 and Theorem A.5, the $embed(D^{init})$ and $prune(P)$ are function-preserving operations and thus there would be no pruning-caused damage if $\|DW\|_{2,1} = 0$.

# B  PROOF AND REMARKS ON THEOREM 3.1

**Theorem 3.1.** *Let $\mathbb{D} = \{D = diag(\delta) | \delta \in \mathbb{R}^{N_W}\}$ be space of diagonal matrices and consider projection map*

$$p : \mathbb{R}^{N_W \times N_I} \times \mathbb{D} \longrightarrow \mathbb{R}^{N_W \times N_I}$$
$$(W, D) \longmapsto W \tag{2}$$

*then*

*(1)  $X_{tgt} = p(\{(W, D) | DW = 0 \text{ and } D \neq 0\})$.*

*(2)  Let $B(X, \epsilon)$ be the $\epsilon$ neighborhood of $X$ with the $L_2$ norm. If $k > 0$ is a positive real number, then*

$$B(X_{tgt}, \epsilon) = p(\{(W, D) | \|DW\|_{2,1} < k\epsilon \text{ and } \|D\|_1 > k\}) \tag{3}$$

*Proof of Theorem 3.1 (1).* We first prove that $X_{tgt} \subseteq p(\{(W, D) | DW = 0 \text{ and } D \neq 0\})$.

if $\overline{W} \in X_{tgt}$, then there exists $\overline{i}$ such that $\overline{W_{\overline{i}}} = 0$. Without loss of generality, let $\overline{i} = 1$ and consider

$$\overline{D} = diag(1, 0, \cdots, 0). \tag{B1}$$

Then $\overline{DW} = 0$ and $\overline{D} \neq 0$. Therefore, we have

$$(\overline{W}, \overline{D}) \in \{(W, D) | DW = 0 \text{ and } D \neq 0\} \tag{B2}$$

and hence

$$\overline{W} = p(\overline{W}, \overline{D}) \in p(\{(W, D) | DW = 0 \text{ and } D \neq 0\}). \tag{B3}$$

For the opposite inclusion, let $\tilde{W} \in p(\{(W, D) | DW = 0 \text{ and } D \neq 0\})$. Then, there exist $\tilde{D} \neq 0$ that satisfy $\tilde{D}\tilde{W} = 0$.

Without loss of generality, we have $\tilde{D}_{11} \neq 0$ and thus $\tilde{W}_1 = 0$ since $(\tilde{D}\tilde{W})_1 = 0$. Therefore, $\tilde{W}$ in $X_{tgt}$ and

$$X = p(\{(W, D) | DW = 0 \text{ and } D \neq 0\}). \tag{B4}$$

□

*Proof of Theorem 3.1 (2).* Suppose $W \in B(X_{tgt}, \epsilon)$ and let $\overline{i} = argmin_{i \in [N_W]} \|W_{i,:}\|_2$. WLOG, let $\overline{i} = 1$ then we have $\|W_{1,:}\|_2 < \epsilon$ and thus we can choose $k' \in (k, \frac{\epsilon}{\|W_{1,:}\|_2} k)$.

Let $\overline{D} = diag(k', 0, \cdots, 0)$. Then we have $\|D\|_1 = k' > k$ and

$$\|DW\|_{2,1} = k' \|W_{1,:}\|_2 + \sum_{j > 1} 0 \cdot \|W_{j,:}\|_2 < k\epsilon. \tag{B5}$$

Therefore, $W \in \{(W, D) | \|DW\|_{2,1} < k\epsilon \text{ and } \|D\|_1 > k\}$.

For the opposite direction, let $\|\tilde{D}\tilde{W}\|_{2,1} < k\epsilon$ and $\|\tilde{D}\|_1 > k$. Let $\tilde{i} = argmin_{i \in [N_W]} \|\tilde{W}_{i,:}\|_2$ Then we have

$$\|\tilde{W}_{\tilde{i},:}\|_2 = \sum_{i=1}^{N_W} \frac{|\tilde{D}_{ii}|}{\|\tilde{D}\|_1} \|\tilde{W}_{\tilde{i},:}\|_2 \leq \sum_{i=1}^{N_W} \frac{|\tilde{D}_{ii}|}{\|\tilde{D}\|_1} \|\tilde{W}_{i,:}\|_2 < \frac{1}{k} \sum_{i=1}^{N_W} |D_{ii}| \|\tilde{W}_{i,:}\|_2 = \frac{1}{k} \|\tilde{D}\tilde{W}\|_{2,1} < \epsilon$$

(B6)

Therefore, we have $\|\tilde{W}_{\tilde{i},:}\|_2 < \epsilon$ and thus $W \in B(X_{tgt}, \epsilon)$.

□

**Remark B.1.**     *1. Since defining equations in $DW$ are homogeneous, the manifold itself becomes affine cone of some projective variety and thus we get invariance under isotropic scaling with any positive real number $k$.*

   *2. Therefore, the absolute value of $\|DW\|$ doesn't mean much in mathematics. We should consider the $\|D\|_1$ together to make it clear whether the elements with small $\|DW\|_{2,1}$ are close to $X_{tgt}$ or not. Although $\|DW\|_{2,1}$ is small enough, if $\|D\|$ is too small then we can't say $W$ is in vicinity of $X_{tgt}$.*

   *3. However, small value of $\|DW\|$ is still important for the removal of $D$, the $proj$ process of bypass algorithm.*

## C    PROOF OF THEOREM A.5

**Theorem A.5.** *Let $prune$ be the mappings defined in Definition A.4. If $DW = 0$ and $P = \{i|D_{ii} \neq 0\} = \{i|\forall jW_{ij} = 0\} \in 2^{N_w}$, then $prune(P)$ becomes function-preserving. That is,*

$$\overline{\varphi_2} = \overline{\varphi_2} \circ prune(P).$$

(A6)

*Proof of Theorem A.5.* Recall that

$$prune(P)(W, b_W, A, b_A, D, \overline{D}) = (W[P^c], b_W[P^c], (A^T[P^c])^T, b'_A, 0, \overline{D}[P^c])$$

(C1)

where $b'_A = b_A + ADb_W + (A^T[P])^T(\psi_{-\overline{D},0}(b_W))[P]$.

Let $|P| = n$ and WLOG let $P = \{1, \cdots, n\}$. Then we can write each parameters can be written by block matrix representations:

$$A = \begin{bmatrix} A^T[P]^T & | & A^T[P^c]^T \end{bmatrix}, W = \begin{bmatrix} 0 \\ W[P^c] \end{bmatrix}, b_W = \begin{bmatrix} b_W[P] \\ b_W[P^c] \end{bmatrix}$$

(C2)

Let $X_{tgt}$ be arbitrary input tensor of $\overline{\varphi_2}(\overline{\theta}, D, \overline{D})$ then the output would be given by

$$\overline{\varphi_2}(\overline{\theta}, D, \overline{D})(x) = b_A + A\psi_{D,\overline{D}}(b_W + Wx)$$

$$= b_A + ADb_W + AD\!\!\!\!\diagup\!\!\!\!W\!x - A\overline{D}b_W - A\overline{D}Wx + A\sigma(b_W + Wx)$$

(C3)

$$= b_A + ADb_W + A\psi_{0,\overline{D}}(b_W + Wx)$$

Since $\psi_{0,\overline{D}}$ is channel-wise operation, $\psi_{-\overline{D},0}(b_W)[P] = \psi_{0,\overline{D}[P]}(b_W[P])$. Considering the block matrix representation Eq. (C2), $A\psi_{0,\overline{D}}(b_W + Wx)$ becomes

$$A\psi_{0,\overline{D}}(b_W + Wx)$$

$$= \begin{bmatrix} A^T[P]^T & | & A^T[P^c]^T \end{bmatrix} \psi_{0,\overline{D}}\left(\begin{bmatrix} b_W[P] \\ b_W[P^c] + W[P^c]x \end{bmatrix}\right)$$

$$= \begin{bmatrix} A^T[P]^T & | & A^T[P^c]^T \end{bmatrix} \begin{bmatrix} \psi_{0,\overline{D}[P]}(b_W[P]) \\ \psi_{0,\overline{D}[P^c]}(b_W[P^c] + W[P^c]x) \end{bmatrix}$$

(C4)

$$= A^T[P]^T \psi_{0,\overline{D}[P]}(b_W[P])$$

$$\qquad + A^T[P^c]^T \psi_{0,\overline{D}[P^c]}(b_W[P^c] + W[P^c]x)$$

Hence, letting $b'_A = b_A + ADb_W + (A^T[P])^T(\psi_{-\overline{D},0}(b_W))[P]$ we finish the proof by

$$
\begin{aligned}
\overline{\varphi_2}(\overline{\theta}, D, \overline{D})(x) &= b'_A + A^T[P^c]^T \psi_{0, \overline{D}[P^c]}(b_W[P^c] + W[P^c]x) \\
&= \overline{\varphi_2}(b_W[P^c], W[P^c], A^T[P^c]^T, b'_A, 0, \overline{D}[P^c]) \\
&= (\overline{\varphi_2} \circ prune)(\overline{\theta}, D, \overline{D})(x)
\end{aligned}
\tag{C5}
$$

$\square$

# D  PROOF OF THEOREM 3.2

**Theorem 3.2.** *Let $d$ be the scalar parameter and $M$ be N-dimensional vector. $M_t$ and $d_t$ be the trajectory of gradient descent movement of $\|dM\|_2$ at timestep $t$ positive learning rate $\lambda_t$ and weight decay term $\alpha \ll 1$. Let $c_t = \frac{|d_t|}{\|M_t\|_2}$ and assume that*

$$
0 < \lambda_* < \lambda_t \ll \min\left(\frac{(1-\alpha)}{c_0}, (1-\alpha)c_0\right)
\tag{6}
$$

*where $\lambda_* = \inf_t \lambda_t$ is the infimum $\lambda_t$.*

*(1) If $c_0 = 1$, then $c_t = 1$ for all timestep $t$.*

*(2) If $c_0 < 1$, then $c_t$ exponentially shrinks to $\frac{\lambda_t}{1-\alpha}$. i.e, If $T$ be the smallest integer satisfying $c_T \le \frac{\lambda_T}{1-\alpha}$ then there exists $k \in (0,1)$ such that $c_{t+1} \le k c_t$ for all $t < T - 1$.*

*(3) If $c_0 > 1$, then $c_t$ exponentially grows to $\frac{(1-\alpha)}{\lambda_t}$. i.e, if $T$ is the smallest integer satisfying $c_T \ge \frac{1-\alpha}{\lambda_T}$ then there exists $k > 1$ such that $c_{t+1} \ge k c_t$ for all $t < T - 1$*

*Proof of Theorem 3.2.* First consider the gradient descent movement of $d_t$ and $M_t^{(i)}$, the $i$th entry of $M_t$, as follows:

$$
d_{t+1} = d_t - \alpha d_t - sgn(d_t)\lambda_t \|M_t\|_2 = (1 - \alpha - \frac{\lambda_t}{c_t})d_t
$$
$$
M_{t+1}^{(i)} = M_t^{(i)} - \alpha M_t^{(i)} - \lambda_t |d_t| \cdot \frac{M_t^{(i)}}{\|M_t\|_2} = (1 - \alpha - \lambda_t c_t)M_t^{(i)}.
\tag{D1}
$$

Note that the second inequalities of each are induced from the definition of $c_t = \frac{|d_t|}{\|M_t\|_2}$.

From second equation of Eq. (D1) we can induce following vector-formed updates:

$$
M_{t+1} = (1 - \alpha - \lambda_t c_t)M_t.
\tag{D2}
$$

Therefore, if

$$
\frac{\lambda_t}{1-\alpha} \le c_t \le \frac{(1-\alpha)}{\lambda_t}
\tag{D3}
$$

then we have

$$
\|M_{t+1}\|_2 = (1 - \alpha - \lambda_t c_t)\|M_t\|_2.
\tag{D4}
$$

and

$$
d_{t+1} = (1 - \alpha - \frac{\lambda_t}{c_t})d_t.
\tag{D5}
$$

Therefore, we get

$$
c_{t+1} = \frac{1 - \alpha - \frac{\lambda_t}{c_t}}{1 - \alpha - \lambda_t c_t} c_t = f(c_t, \lambda_t)c_t
\tag{D6}
$$

where $f(x, y) = \frac{1 - \alpha - \frac{y}{x}}{1 - \alpha - xy}$.

(1) Suppose $c_0 = 1$ then simple induction shows that $c_t = 1$ for all time $t$ since Eq. (D3) holds by assumption in Eq. (6).

(2) Suppose $c_0 < 1$ and let $T$ be the smallest integer which satisfies $\frac{\lambda_T}{1-\alpha} > c_T$.

If $t < T$ and $c_t < 1$, then $f(c_t, \lambda_t) < 1$ by Lemma D.1(1) and thus $c_{t+1} < c_t < 1$, which means that $\{c_t\}$ is a decreasing sequence.

Also, by Lemma D.1(1) we have $f(c_t, \lambda_t) < f(c_0, \lambda_*) < 1$ since $c_t < c_0$ because $\{c_t\}$ is a decreasing sequence, and $\lambda_t > \lambda_*$ due to the assumption in Eq. (6). Therefore, we have

$$c_t < f(c_0, \lambda_*)c_{t-1} < \cdots < f(c_0, \lambda_*)^t c_0 \tag{D7}$$

which finishes the proof of (2).

(3) Suppose $c_0 > 1$ and let $T$ be the smallest integer which satisfies $\frac{\lambda_T}{1-\alpha} < c_T$.

If $t < T$ and $c_t > 1$, then $f(c_t, \lambda_t) > 1$ by Lemma D.1(2) and thus $c_{t+1} > c_t > 1$, which means that $\{c_t\}$ is a increasing sequence.

Also, by Lemma D.1(2) we have $f(c_t, \lambda_t) > f(c_0, \lambda_*) > 1$ since $c_t > c_0$ because $\{c_t\}$ is a increasing sequence, and $\lambda_t > \lambda_*$ due to the assumption in Eq. (6). Therefore, we have

$$c_t > f(c_0, \lambda_*)c_{t-1} > \cdots > f(c_0, \lambda_*)^t c_0 \tag{D8}$$

which completes the proof of (3).

$\square$

**Lemma D.1.** *Let* $f(x, y) = \frac{1 - \alpha - \frac{y}{x}}{1 - \alpha - xy}$ *and* $y \in (0, 1 - \alpha)$.

*(1) If* $x \in (0, 1)$ *then* $f(x, y) < 1$, $\frac{\partial f}{\partial x} > 0$ *and* $\frac{\partial f}{\partial y} < 0$

*(2) If* $x \in (1, \infty)$ *then* $f(x, y) > 1$, $\frac{\partial f}{\partial x} > 0$ *and* $\frac{\partial f}{\partial y} > 0$

*Proof.* We first compute the partial derivatives:

$$\frac{\partial f}{\partial x} = \frac{y(1 - \alpha)}{x^2(1 - \alpha - xy)^2}\{(x - \frac{y}{1 - \alpha})^2 - \frac{y^2}{(1 - \alpha)^2} + 1\} \tag{D9}$$

$$\frac{\partial f}{\partial y} = \frac{(1 - \alpha)(x^2 - 1)}{x(1 - \alpha - 1xy)^2} \tag{D10}$$

Since $y \in (0, 1 - \alpha)$, the $-\frac{y^2}{(1-\alpha)^2} + 1$ term in RHS of Eq. (D9) becomes positive. Therefore, $\frac{\partial f}{\partial x} > 0$ for all $X_{tgt}$.

Now assume that $x \in (0, 1)$. Then $f(x, y) < 1$ since $x < \frac{1}{x}$ and $\frac{\partial f}{\partial y} < 0$ because $x^2 - 1 < 0$.

Similarly, if $x \in (1, \infty)$ then $f(x, y) > 1$ and $\frac{\partial f}{\partial y} > 0$. $\square$

# E    DETAILS ON EXPERIMENTS

In this section, we list the details on expeirment designs.

For CIFAR experiments, the original model is not fixed along the benchmark in literature, which may raise reproducibility issue. To mitigate this, we exploit pretrained model provided by Fang et al. (2023) which achieve 93.53% and 73.50% top-1 accuracy, respectively, to get comparable baseline sharing the same starting point. In Table 2, only DepGraph results are under fair comparison.

For the Imagenet experiments, to show only the effect of our method, we follow the strict principle that no extra engineering skills such as additional augmentation, label smoothing, mixup, distillation, etc. are, unless if they are used in official recipe of pretraining, provided by TorchVision(maintainers & contributors, 2016). The benchmark models using extra skills (and compared to models trained with vanilla settings) were not included in Table 2 for the fair comparison.

All experiments in this paper were conducted with Linux (Ubuntu 20.04) computer equipped with single RTX 4090 GPU with 24GB VRAM. For imagenet experiment, we use gradient accumulation to run with single GPU. The hyperparmeters used in each expriments are summarized in following tables.

| | Resnet56+CIFAR10 | VGG19+CIFAR100 |
|---|---|---|
| batch size | 128 | 128 |
| Gradient accumulation | 1 | 1 |
| optimizer | SGD(momentum=0.9) | SGD(momentum=0.9) |
| Weight decay $(\alpha_\theta, \alpha_D)$ | (5e-4,5e-5) | (5e-4,0) |
| c | 1 | 1 |
| $\gamma_t$ | 0.018(1+t/4), | $\gamma$(1+0.25t), $\gamma$=[2e-3,9e-3,12e-3] |
| $\epsilon$ (opt1,opt2) | 5e-7,1e-6 | 2e-6,1e-6 |
| Stage finish epochs (opt1,opt2,finetune) | 50,100,200 | 200,300,400 |
| LR (opt1,opt2,finetune) | 1e-2,1e-2,5e-3 | 5e-3,5e-3, 1e-3 for 3x
0.01, 0.01, 0.01 |
| LR decay epoch (opt1,opt2,finetune) | [25,40],[75,90],[120,150,170,190] | [],[],[390] for 3x
[75,750],[250],[340,370] for rest |
| LR decay ratio (opt1,opt2,finetune) | 0.1, 0.1, 0.1 | NA,NA,0.1 for 3x
0.2,0.2,0.1 for rest |
| training runtime | 1.6 hours | 2 hours |

| | Resnet50+Imagenet | DeiT-Tiny+Imagenet |
|---|---|---|
| batch size | 64 | 256 |
| Gradient accumulation | 2 (SL), 1*(4 GPUs) (FT) | 4 |
| optimizer | SGD(momentum=0.9) | Adamw(0.9,0.99) |
| Weight decay $\alpha_\theta$) (opt1,opt2,finetune) | 1e-4,1e-4,2e-5 | 0.05,0.05,0.05 |
| Weight decay $\alpha_D$ | 0 | 0.05 |
| c | 1 | 0.55 for mlp, 0.2 for attention |
| $\gamma_t$ | 3e-4(1+0.25t) | 5e-3(1000+t/4) |
| $\epsilon$ (opt1,opt2) | 3e-6,3e-6 | 3e-6,3e-6 (pruning off at opt2) |
| Stage finish epochs (opt1,opt2,finetune) | 20,40,140 | 20,42,350 |
| LR (opt1,opt2,finetune) | 1e-4,1e-4,0.2 | 2.5e-4,2.5e-4,0.001 |
| LR decay epoch (opt1,opt2,finetune) | [5,10],[25,40],[70,90,125] | 20,40,350 |
| LR decay ratio (opt1,opt2,finetune) | 0.1,0.1,0.1 | cosine decay(min_lr=1e-5) |
| training runtime | 44 (SL)+23 (FT,4GPU) hours | 106.75hours |
| mixup alpha | N/A | 0.2 |
| cutmix alpha | N/A | 1 |
| label smoothing | N/A | 0.1 |
| mixup prob | N/A | 1 |
| ColorJitter | N/A | 0.3 |
| Auto augment | N/A | rand-m9-mstd0.5-inc1 |

# F LOSS CURVES DURING THE TRAINING

In this section, we plot the training logs of loss, accuracy, $\|DW\|$ and MACs, in CIFAR10 experiment. The target layers of the model were pruned early in epoch 51 and 116.

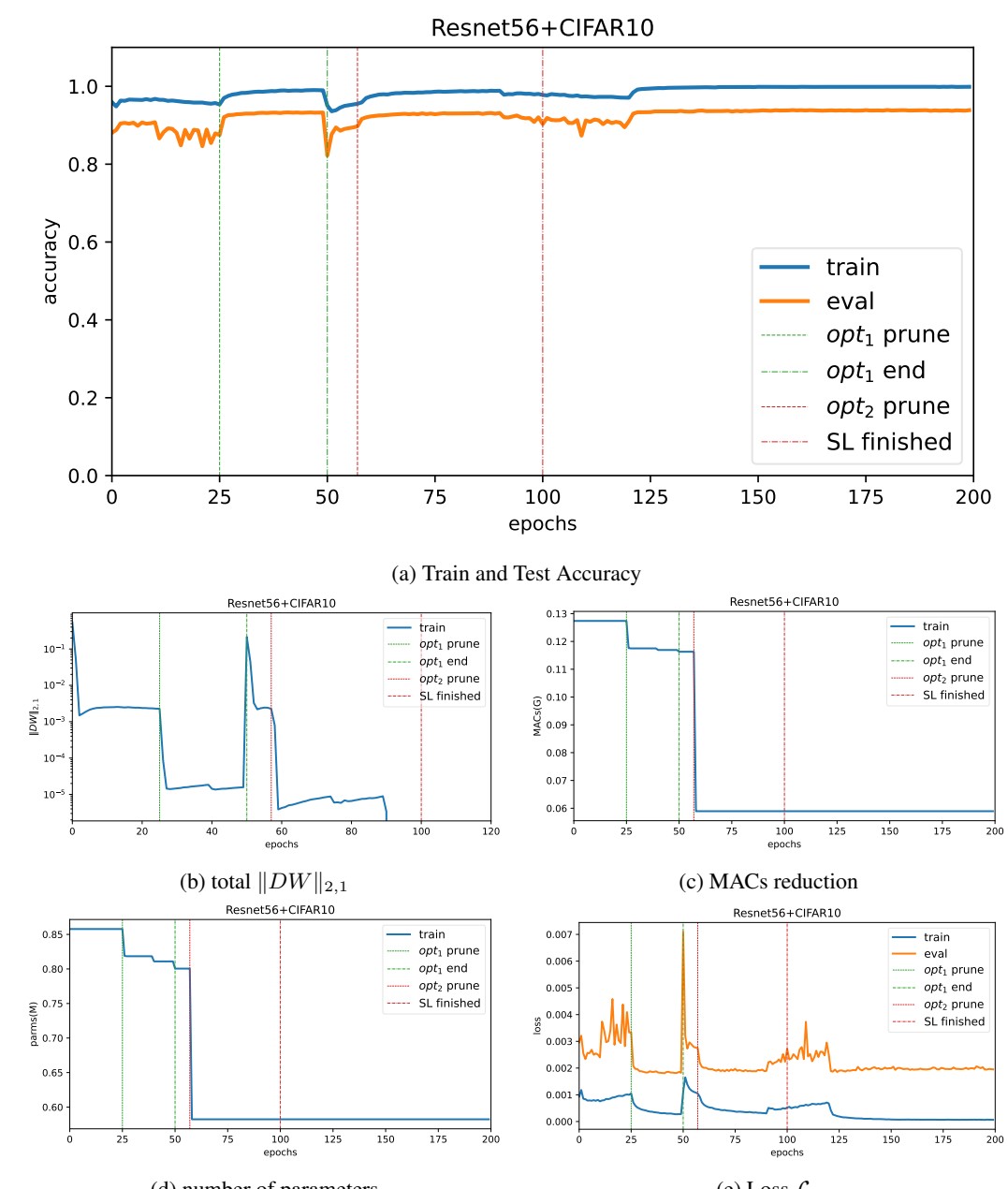

(a) Train and Test Accuracy

(b) total $\|DW\|_{2,1}$

(c) MACs reduction

(d) number of parameters

(e) Loss $\mathcal{L}_2$

Figure 5: Learning curves of CIFAR10 experiment with Resnet56

# G IMPLEMENTATION DETAILS FOR ATTENTION

Attention layer of transformer, mostly the MHA (Multi-Head self attention) requires several modifications to prune in sense NP (neuron pruning) which introduced in (Shim et al., 2024).

First, the MHA does not include any activation layer $\sigma$ inside the attention score computation, thus we consider the auxiliary identity activation right after the QKV computation and extend the model by replacing this identity function $\psi_D(x) = Dx + x$. In initialization of D, we consider the multiplier $m_i$ which is solution of quadratic equation $\frac{1}{m_i} - 1 = m_i c_0 \|F_i\|$ for each filter index $i$. After, we rescale $F_i$ by $F_i \leftarrow m_i F_i$ and initialize $D_{ii} \leftarrow \frac{1}{m_i} - 1$. Setting this initialization as $embed$, it becomes function-preserving and place the $(D_i i, F_i)$ to the same decision boundary explained in Section 3.3.

Second, due to latency issue, the number of neurons of each head must be equal for the parallel operation in GPU. Therefore, we group the neurons to be regularized and pruned together. After grouping, we minimize the multiplication of each concatenated $F_i$ and $D_{ii}$ vector's norm.

Lastly, since Q and K must have equal dimension, we take union of set of pruning indices and prune corresponding neurons together.

# H FULL EVALUATION ON BIFURCATION BEHAVIOR

In this section, we provide histograms of filter norm, $\{D_{ii}\}_{i\in[N_W]}$ and $C = \frac{D}{\|W\|_{2,1}}$ for every layer of our pruned VGG19 model (in Section 4.6) and Resnet50 model(speedup 2.00x), to show that the bifurcation behavior claimed in Section 3.3 always happens.

## H.1 RESNET56+CIFAR10

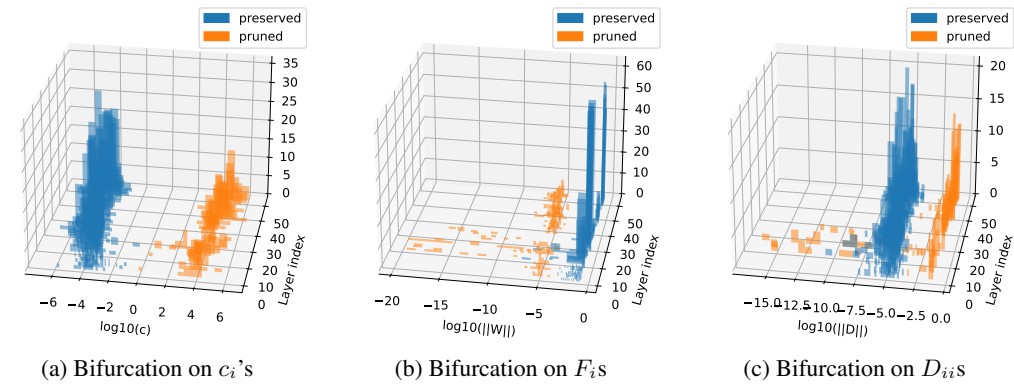

(a) Bifurcation on $c_i$'s     (b) Bifurcation on $F_i$s     (c) Bifurcation on $D_{ii}$s

Figure 6: The histograms of ratio $c = \frac{D_{ii}}{\|F_i\|}$, filter vector $F_i$ of weight tensor $W$, and $D_{ii}$'s for each layers of Resnet56 model trained on CIFAR10. The z-axis represents the frequency. The layer indices of results in $opt2$ are shifted to start from the last layer.
.

## H.2 VGG19+CIFAR100

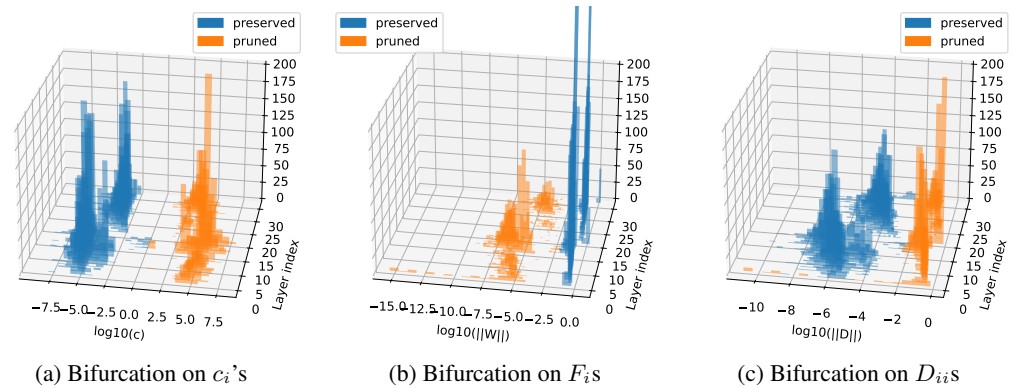

(a) Bifurcation on $c_i$'s     (b) Bifurcation on $F_i$s     (c) Bifurcation on $D_{ii}$s

Figure 7: The histograms of ratio $c = \frac{D_{ii}}{\|F_i\|}$, filter vector $F_i$ of weight tensor $W$, and $D_{ii}$'s for each layers of VGG19 model (3x speedup) trained on CIFAR100. The z-axis represents the frequency. The layer indices of results in $opt2$ are shifted to start from the last layer.
.

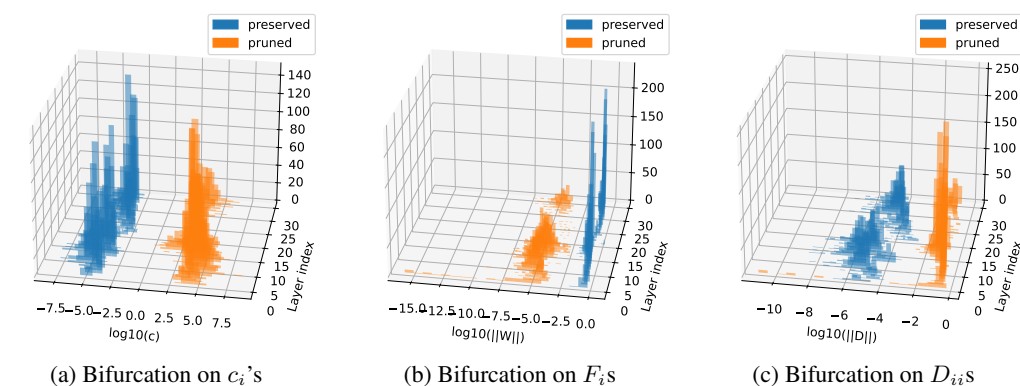

(a) Bifurcation on $c_i$'s  (b) Bifurcation on $F_i$s  (c) Bifurcation on $D_{ii}$s

Figure 8: The histograms of ratio $c = \frac{D_{ii}}{\|F_i\|}$, filter vector $F_i$ of weight tensor $W$, and $D_{ii}$'s for each layers of VGG19 model (8.96x speedup) trained on CIFAR100. The z-axis represents the frequency. The layer indices of results in $opt2$ are shifted to start from the last layer.
.

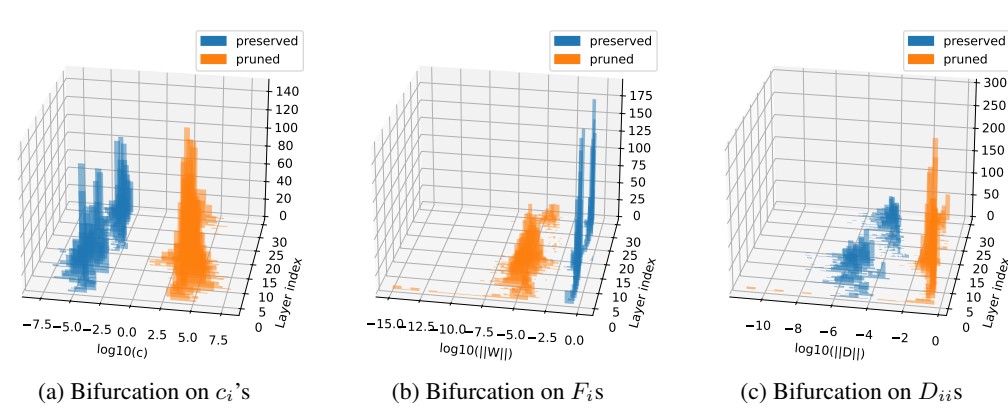

(a) Bifurcation on $c_i$'s  (b) Bifurcation on $F_i$s  (c) Bifurcation on $D_{ii}$s

Figure 9: The histograms of ratio $c = \frac{D_{ii}}{\|F_i\|}$, filter vector $F_i$ of weight tensor $W$, and $D_{ii}$'s for each layers of VGG19 model (11.84x speedup) trained on CIFAR100. The z-axis represents the frequency. The layer indices of results in $opt2$ are shifted to start from the last layer.
.

## H.3 RESNET50+IMAGENET

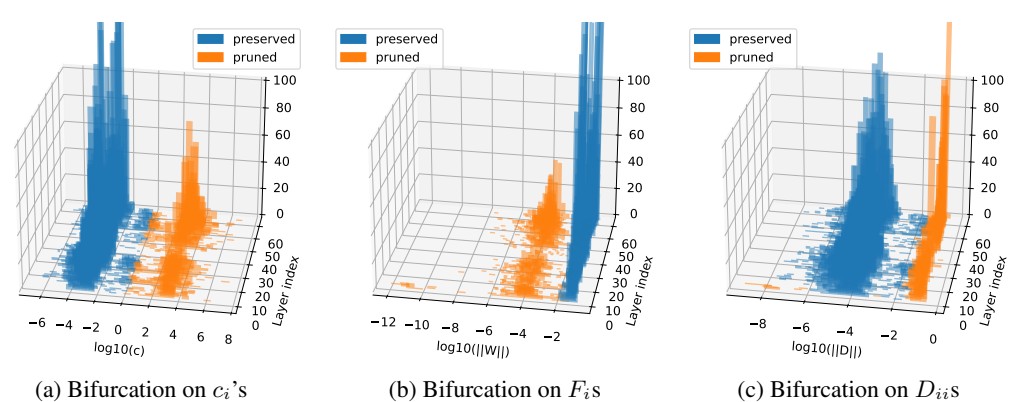

(a) Bifurcation on $c_i$'s    (b) Bifurcation on $F_i$s    (c) Bifurcation on $D_{ii}$s

Figure 10: The histograms of ratio $c = \frac{D_{ii}}{\|F_i\|}$, filter vector $F_i$ of weight tensor $W$, and $D_{ii}$'s for each layers of Resnet50 model (1.49x speedup) trained on Imagenet. The z-axis represents the frequency. The layer indices of results in $opt2$ are shifted to start from the last layer.

.

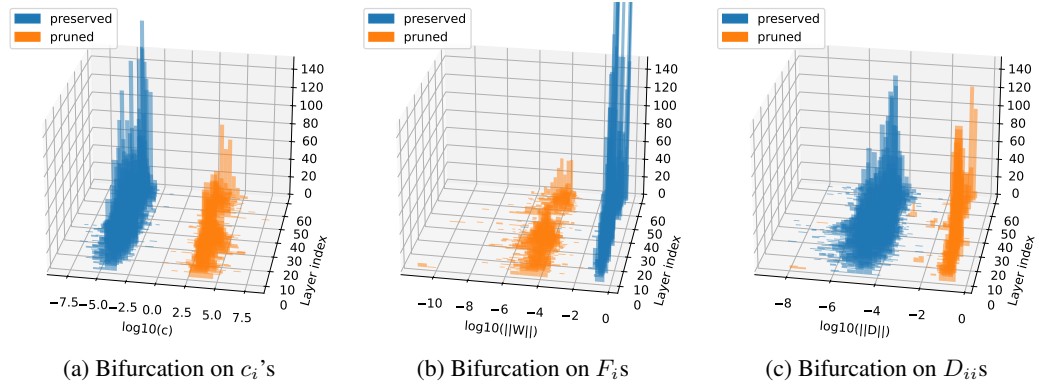

(a) Bifurcation on $c_i$'s    (b) Bifurcation on $F_i$s    (c) Bifurcation on $D_{ii}$s

Figure 11: The histograms of ratio $c = \frac{D_{ii}}{\|F_i\|}$, filter vector $F_i$ of weight tensor $W$, and $D_{ii}$'s for each layers of Resnet50 model (2.18x speedup) trained on Imagenet. The z-axis represents the frequency. The layer indices of results in $opt2$ are shifted to start from the last layer.

.

# I VISUALIZATIONS ON FAIR PRUNING CHANCE

The L1 and Group Lasso regularizer is claimed to prefer the filters with small initial magnitude. In this section, we regularize VGG19 model on CIFAR100, prune filters according to magnitude with pruning ratio which is same to our pruned VGG19 model (speedup 8.96x in Table 2) and visualize the initial magnitude of pruned filters on every layers.

We also visualize the histogram of filter norms after the regularization too. The bifurcation behaviors can be found on some layers, but still there is preference of small initial magnitude.

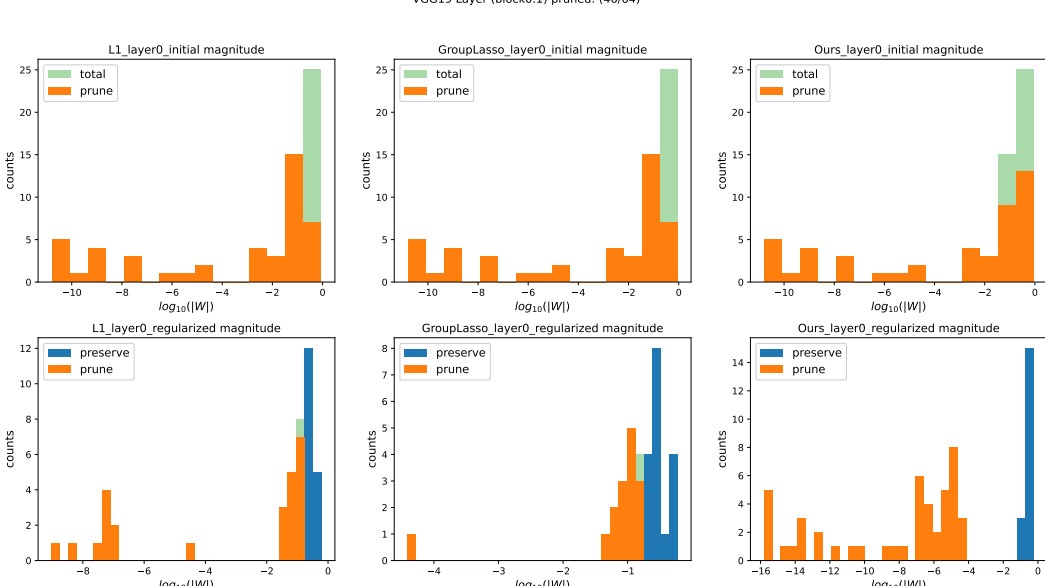

Figure 12: Initial magnitude and regularized magnitude of L1, Group Lasso and our regularizer on 1st layer of VGG19 model. Each columns correspond to L1, Group Lasso, $\|DW\|_{2,1}$ respectively. First row is histogram of initial filter norms and second is historgram of regularized filter norms.

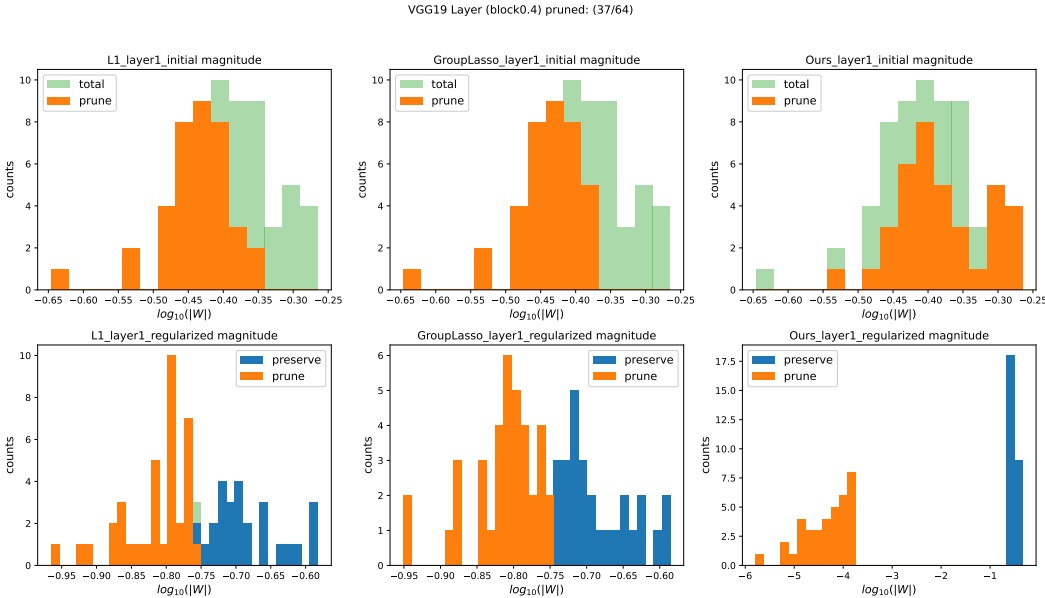

Figure 13: Initial magnitude and regularized magnitude of L1, Group Lasso and our regularizer on 2nd layer of VGG19 model. Each columns correspond to L1, Group Lasso, $\|DW\|_{2,1}$ respectively. First row is histogram of initial filter norms and second is historgram of regularized filter norms.

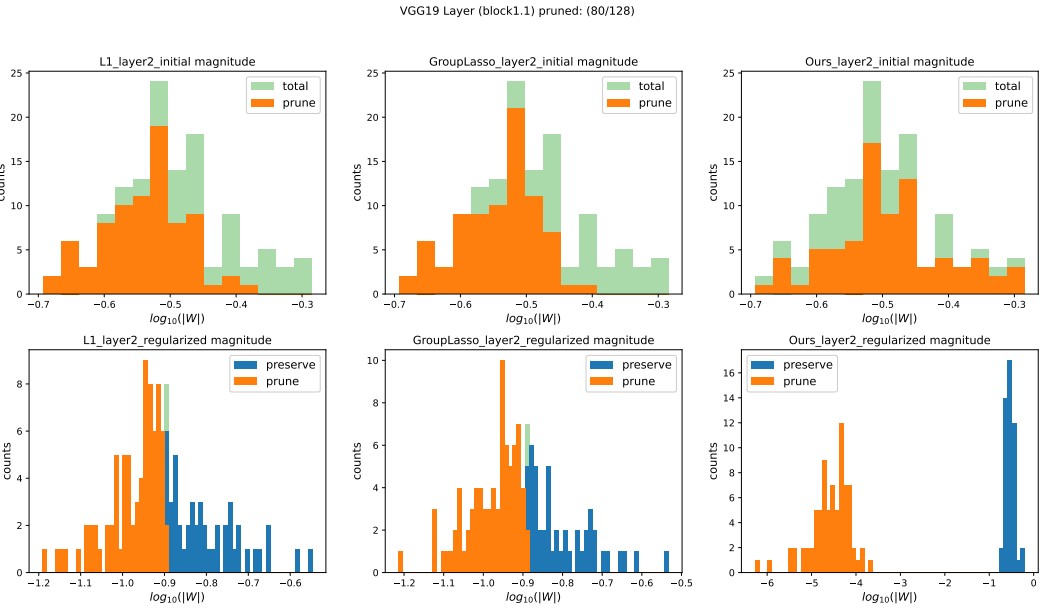

Figure 14: Initial magnitude and regularized magnitude of L1, Group Lasso and our regularizer on 3rd layer of VGG19 model. Each columns correspond to L1, Group Lasso, $\|DW\|_{2,1}$ respectively. First row is histogram of initial filter norms and second is historgram of regularized filter norms.

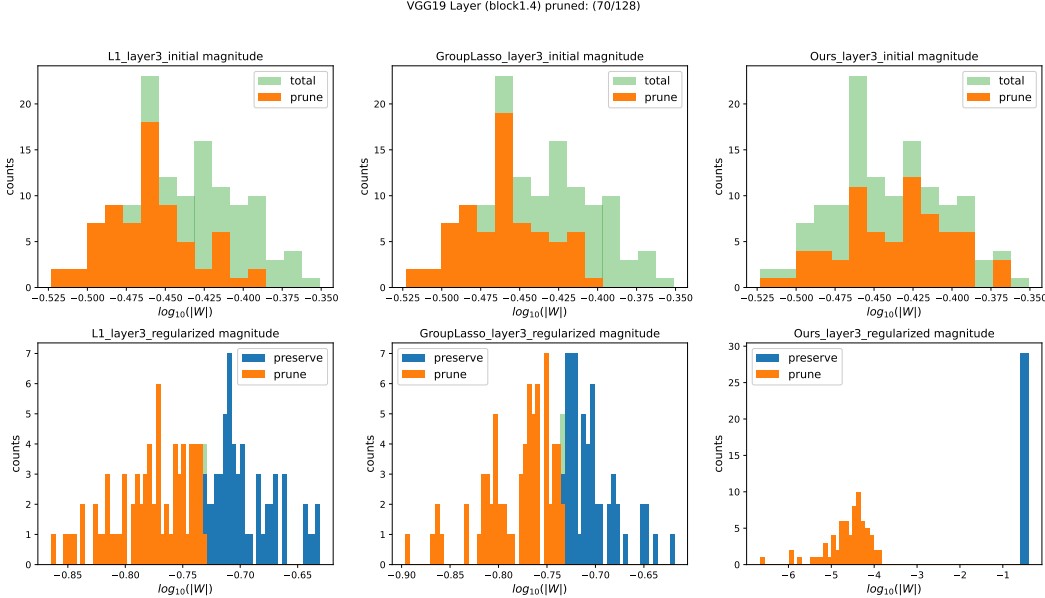

Figure 15: Initial magnitude and regularized magnitude of L1, Group Lasso and our regularizer on 4th layer of VGG19 model. Each columns correspond to L1, Group Lasso, $\|DW\|_{2,1}$ respectively. First row is histogram of initial filter norms and second is historgram of regularized filter norms.

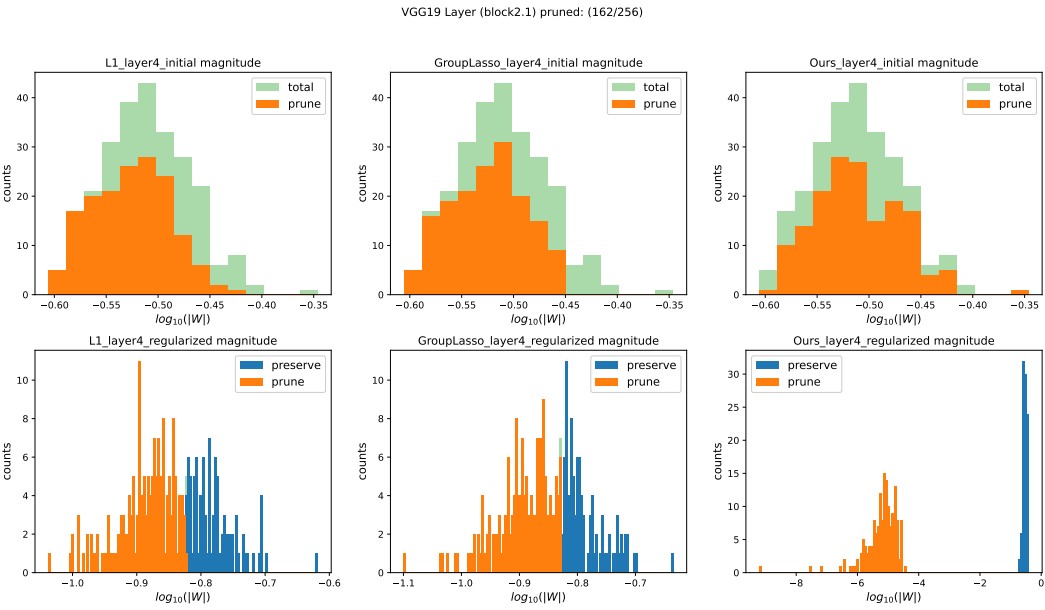

Figure 16: Initial magnitude and regularized magnitude of L1, Group Lasso and our regularizer on 5th layer of VGG19 model. Each columns correspond to L1, Group Lasso, $\|DW\|_{2,1}$ respectively. First row is histogram of initial filter norms and second is historgram of regularized filter norms.

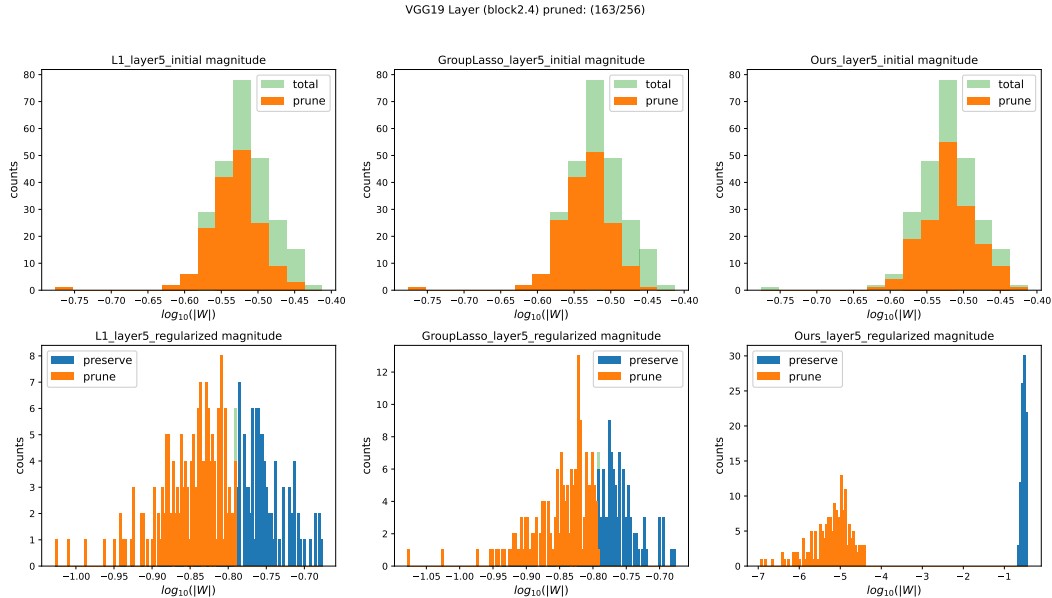

Figure 17: Initial magnitude and regularized magnitude of L1, Group Lasso and our regularizer on 6th layer of VGG19 model. Each columns correspond to L1, Group Lasso, $\|DW\|_{2,1}$ respectively. First row is histogram of initial filter norms and second is historgram of regularized filter norms.

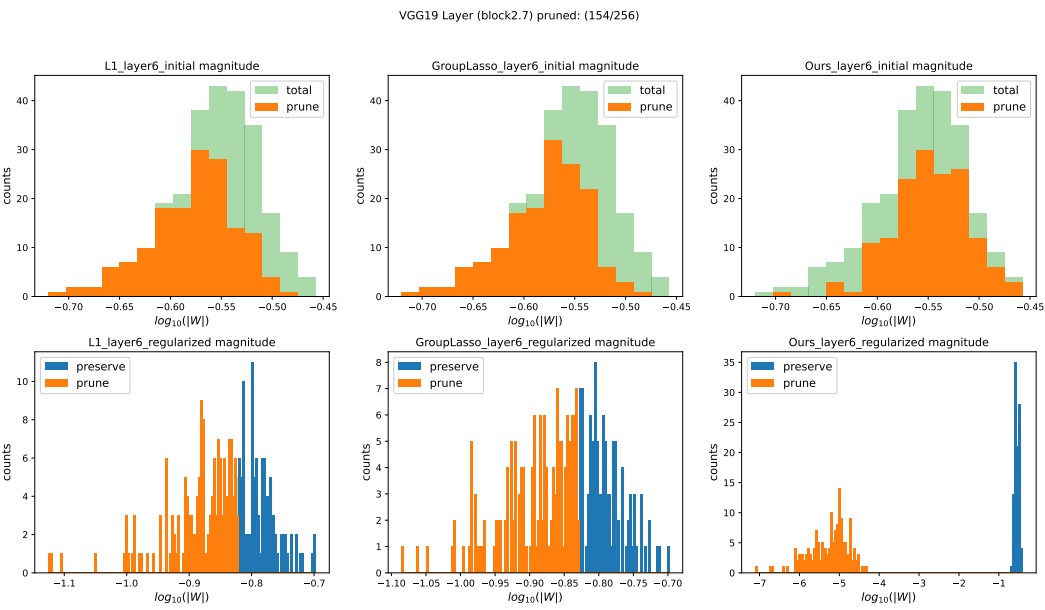

Figure 18: Initial magnitude and regularized magnitude of L1, Group Lasso and our regularizer on 7th layer of VGG19 model. Each columns correspond to L1, Group Lasso, $\|DW\|_{2,1}$ respectively. First row is histogram of initial filter norms and second is historgram of regularized filter norms.

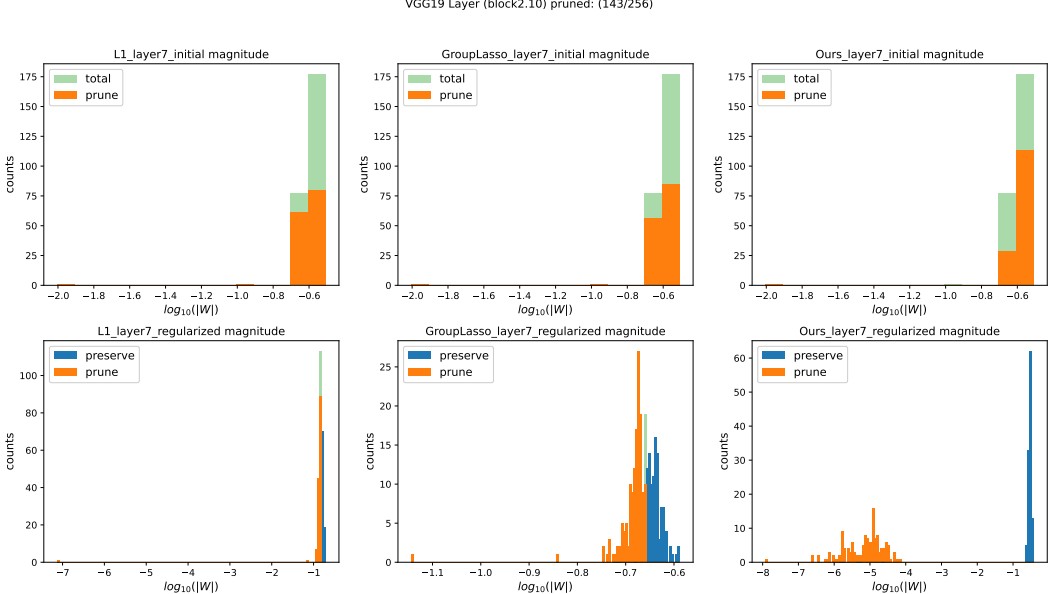

Figure 19: Initial magnitude and regularized magnitude of L1, Group Lasso and our regularizer on 8th layer of VGG19 model. Each columns correspond to L1, Group Lasso, $\|DW\|_{2,1}$ respectively. First row is histogram of initial filter norms and second is historgram of regularized filter norms.

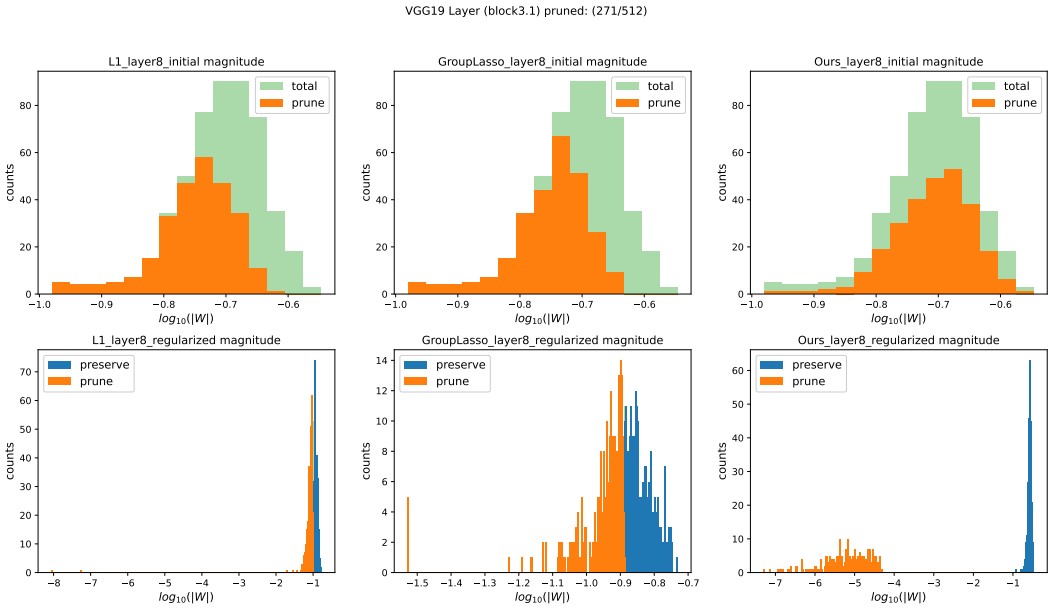

Figure 20: Initial magnitude and regularized magnitude of L1, Group Lasso and our regularizer on 9th layer of VGG19 model. Each columns correspond to L1, Group Lasso, $\|DW\|_{2,1}$ respectively. First row is histogram of initial filter norms and second is historgram of regularized filter norms.

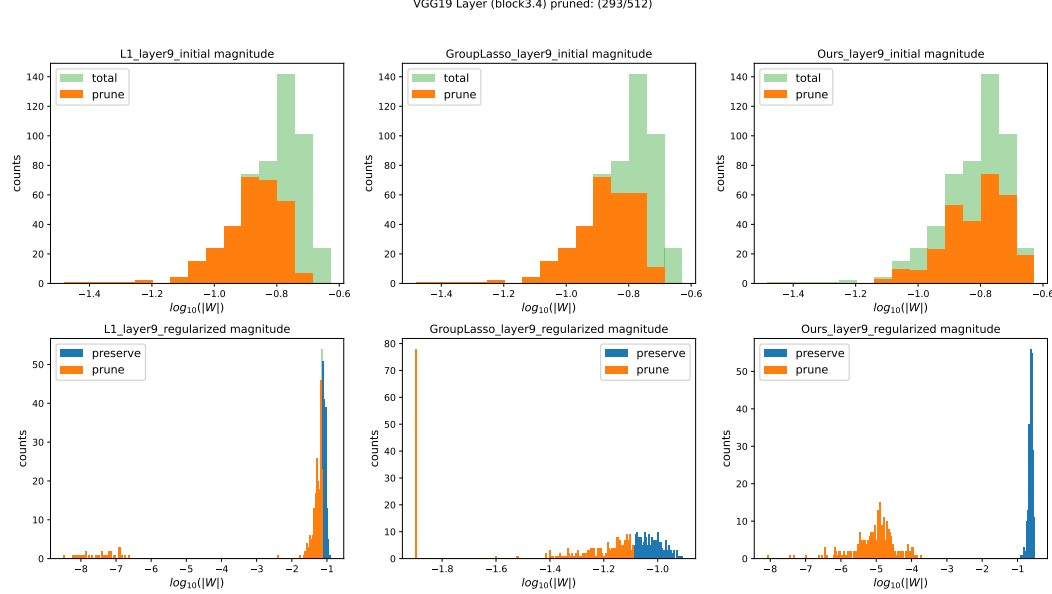

Figure 21: Initial magnitude and regularized magnitude of L1, Group Lasso and our regularizer on 10th layer of VGG19 model. Each columns correspond to L1, Group Lasso, $\|DW\|_{2,1}$ respectively. First row is histogram of initial filter norms and second is historgram of regularized filter norms.

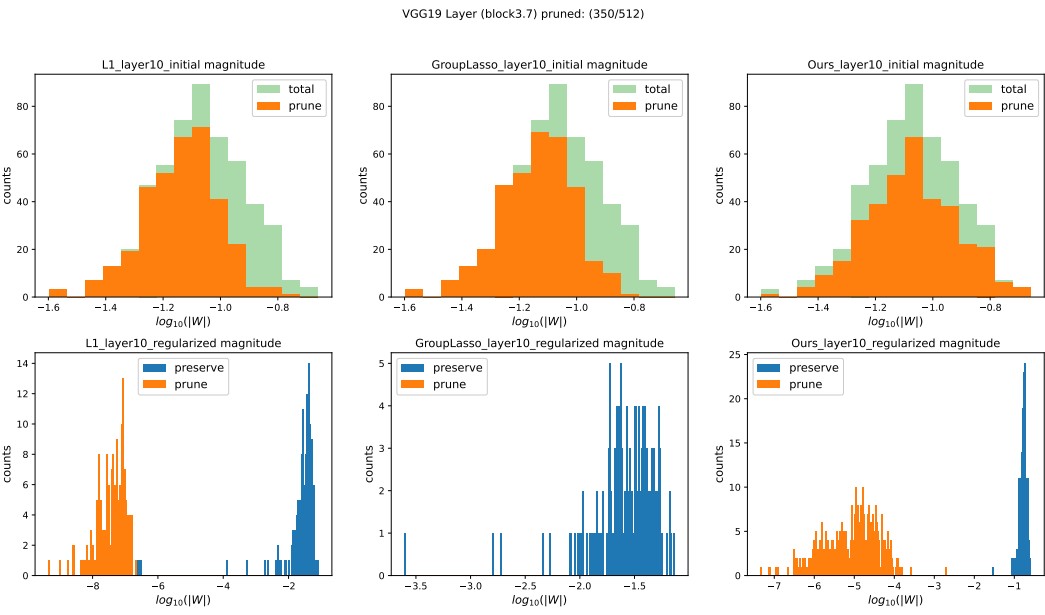

Figure 22: Initial magnitude and regularized magnitude of L1, Group Lasso and our regularizer on 11th layer of VGG19 model. Each columns correspond to L1, Group Lasso, $\|DW\|_{2,1}$ respectively. First row is histogram of initial filter norms and second is historgram of regularized filter norms.

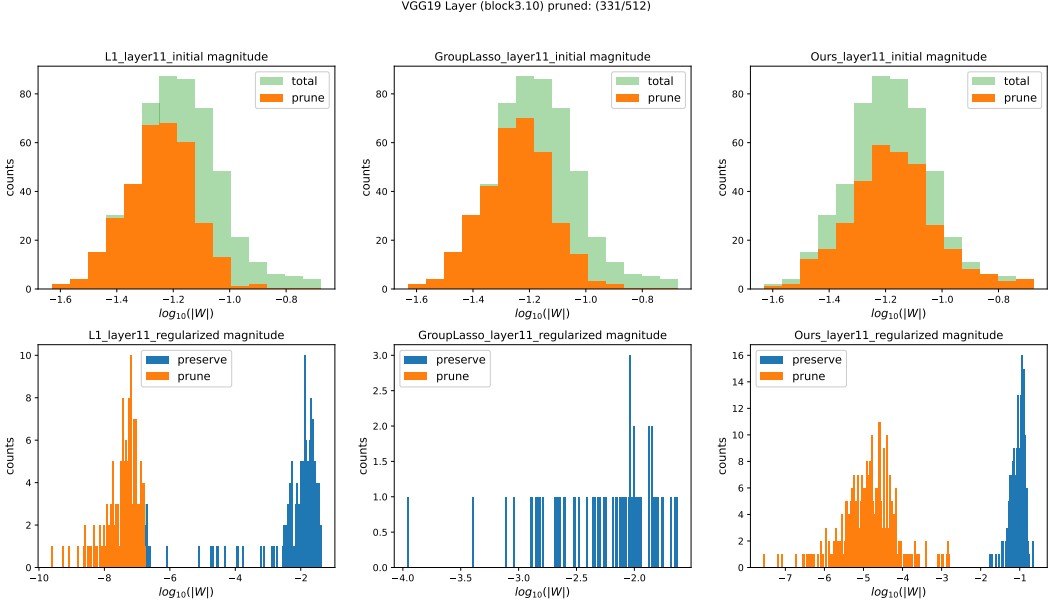

Figure 23: Initial magnitude and regularized magnitude of L1, Group Lasso and our regularizer on 12th layer of VGG19 model. Each columns correspond to L1, Group Lasso, $\|DW\|_{2,1}$ respectively. First row is histogram of initial filter norms and second is historgram of regularized filter norms.

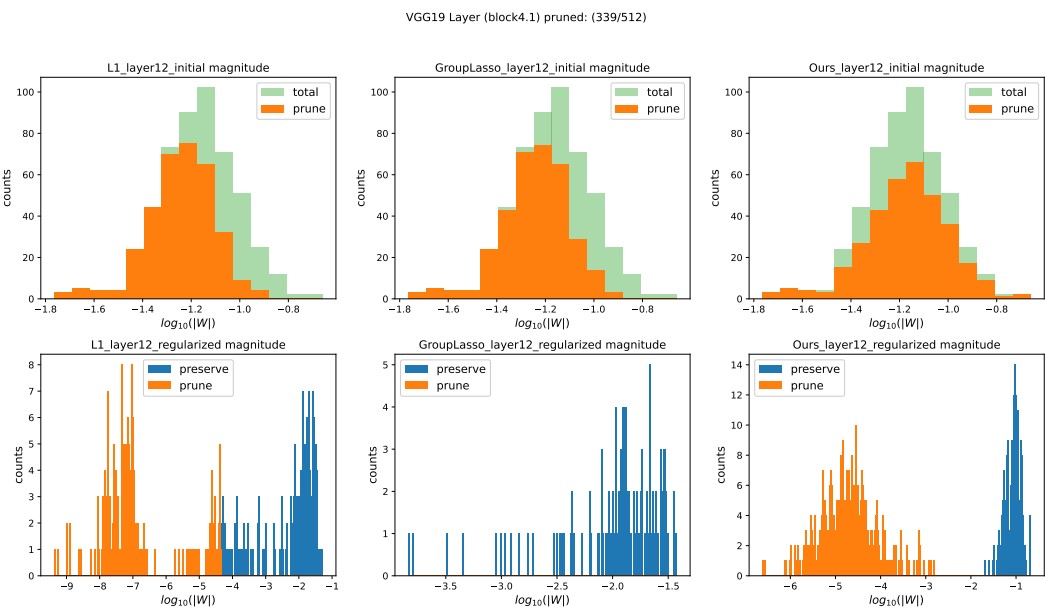

Figure 24: Initial magnitude and regularized magnitude of L1, Group Lasso and our regularizer on 13rd layer of VGG19 model. Each columns correspond to L1, Group Lasso, $\|DW\|_{2,1}$ respectively. First row is histogram of initial filter norms and second is historgram of regularized filter norms.

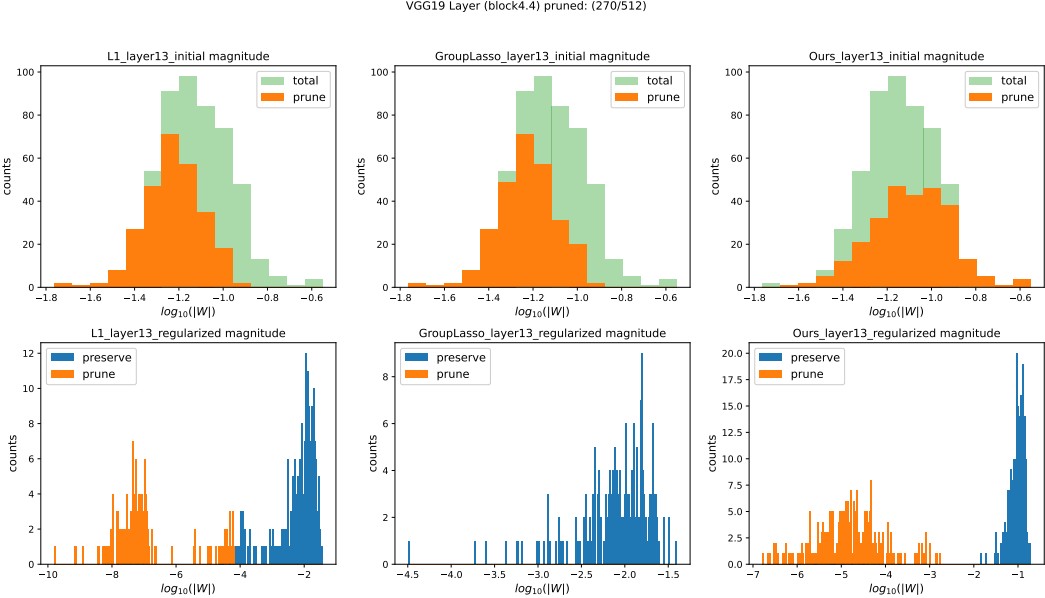

Figure 25: Initial magnitude and regularized magnitude of L1, Group Lasso and our regularizer on 14th layer of VGG19 model. Each columns correspond to L1, Group Lasso, $\|DW\|_{2,1}$ respectively. First row is histogram of initial filter norms and second is historgram of regularized filter norms.

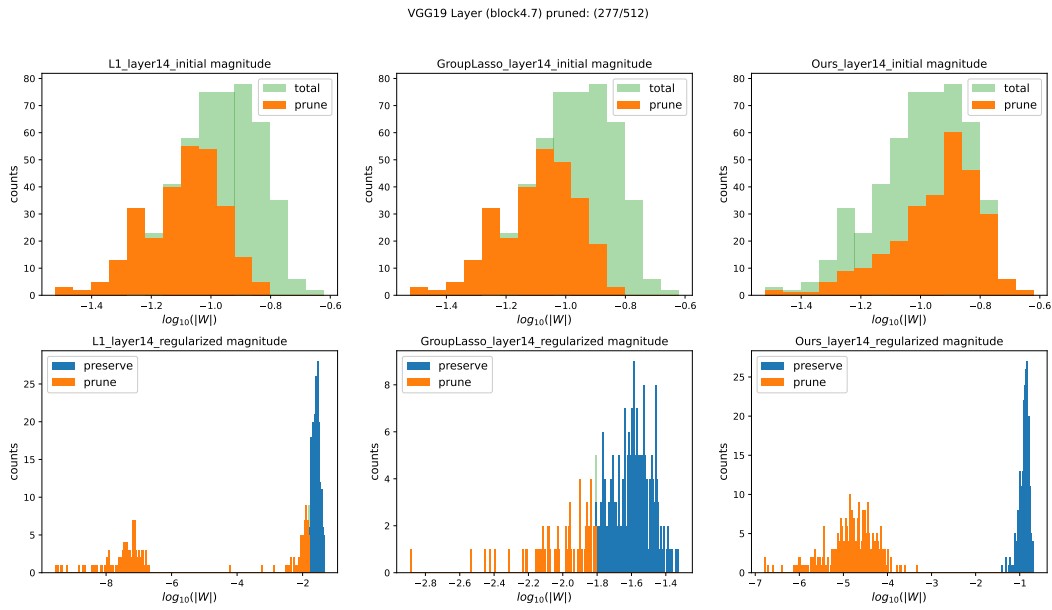

Figure 26: Initial magnitude and regularized magnitude of L1, Group Lasso and our regularizer on 15th layer of VGG19 model. Each columns correspond to L1, Group Lasso, $\|DW\|_{2,1}$ respectively. First row is histogram of initial filter norms and second is historgram of regularized filter norms.

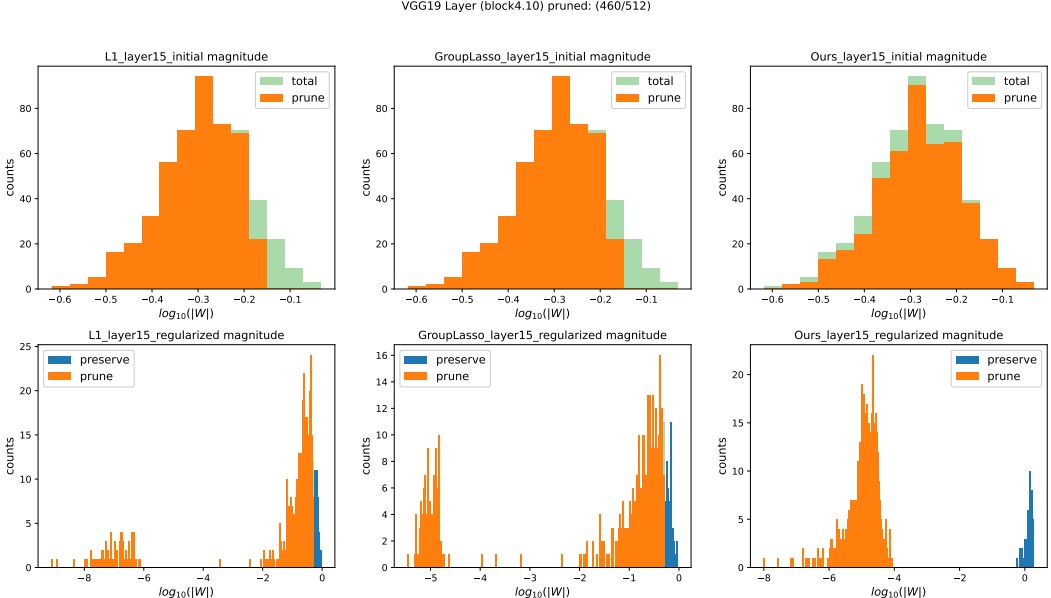

Figure 27: Initial magnitude and regularized magnitude of L1, Group Lasso and our regularizer on 16th layer of VGG19 model. Each columns correspond to L1, Group Lasso, $\|DW\|_{2,1}$ respectively. First row is histogram of initial filter norms and second is historgram of regularized filter norms.

