# OpenReview forum: "Catalyst: Reveal the Geometry of Pruning by Reshaping Neural Network"
_ICLR.cc/2026/Conference — Submitted to ICLR 2026_

### Official Review · Reviewer_mLQT · 2025-10-28

**Soundness:** 2
**Presentation:** 3
**Contribution:** 2
**Rating:** 2
**Confidence:** 4

**Summary:**

Motivated by the weakness of L1 regularization and group lasso, this paper characterizes the lossless pruning via geometry and proposes a regularization object whose global minimum is non-trivial and corresponds to lossless pruning. They propose catalyst pruning based on the bypassing algo.  The model will be pruned after $\|DW\|_{2,1}$ is small enough. Experiments show that the training of catalyst pruning is more smooth than L1/Group Lasso, and the performance of fine-tuning outperforms some existing methods.

**Strengths:**

1. The clear algebraic conditions for "lossless pruning" are given from a geometric perspective ($DW=0$) and the regularization construction of the extended space ($\|DW\|_{2,1}$), with novel ideas and clear expressions.

2. $c=D_{ii}/\|F\|$ naturally results in pruning decisions with wide intervals and no bias; the results are competitive with some existing methods.

**Weaknesses:**

1. The method is “lossless” only at the pruning step, whatever the test acc is. In Table 1, several settings still experience sizeable accuracy recovery at finetuning stage, e.g., ResNet-50 on ImageNet shows post-prune accuracy of 65.13 and finetuning acc of 76.04; DeiT-Tiny also exhibits 62.97 before recovery(71.41). This suggests the lossless guarantee is fragile in practice, the algorithm of the paper can indeed ensure that after training, pruning will not cause loss degradation, but the performance at this time cannot be guaranteed (for example, I can prune at a low performance, and lossless is relatively easy to achieve at this level). Retraining is still required to recover performance matching other algorithms. Therefore, although the theoretical part of the paper gives seemingly powerful results, its actual results not seem to require lossless, as the finetuning stage is necessary. Thus, another question arises: Is lossless pruning necessary? It is obvious that even if the loss is damaged, I can recover it through finetuning.

2. I have some questions about motivation, the paper is stated to address two problems, one is that L1/Group Lasso paradigms are small but important filters are pruned and the other is that decision boundaries are fragile. Is the former a flaw specific to L1/Group Lasso? If so, can addressing this flaw do something that other algorithms can't? As far as I know, there are many algorithms for structured pruning, and L1/Group Lasso is not the only solution. I appreciate the author's efforts on the problem of fragile decision boundaries, so I would like to get more explanations from the author about motivation.

3. I'm very curious about the choice of hyperparameters, the hyperparameters of common algorithms generally guarantee performance after training, the paper claims that their regularization is harmless to performance, but the experimental results prove that the performance of the trained model decreases significantly, and finetuning is needed to recover it. This indicates nontrivial sensitivity to hyperparameters and compute budget, but a systematic robustness study is missing.

4. For DeiT-Tiny, the paper restricts comparisons to _neuron-level_ pruning and explicitly notes that other prunable targets (embed dimension, attention heads) are left out. This narrows applicability for transformers, where head/token/embed pruning is common. Moreover, comparisons are grouped by “similar speedups,” but FLOPs/params/latency trade-offs and hardware-aware speed are not jointly normalized or reported as alternative axes, which could bias comparisons across methods with different effects.

5. No generalization/error-propagation bounds linking $\|DW\|_{2,1}$​ to post-prune risk, this weakness is related to weakness 1.

6. The method argues that decisions are independent of initial filter magnitudes. I'm wondering that is this effect robust under reparameterization? A notable problem with magnitude pruning is that magnitude can be changed significantly by reparameterization while without changing its importance, which is the core reason for the shortcomings of the L1/Group Lasso, did the paper's algorithm solve this problem? Clarifying issues like this in the text would be a significant improvement in quality.

7. The algorithms compared in the paper are still not comprehensive enough, as far as I know there are many powerful algorithms, such as HRank, CHIP, FPAC, HBFP, SPSRC, HALP, MDP, IPPRO, etc. Since there are many algorithms, I have not made a complete list, I would like to ask the authors to choose the algorithms according to the actual to supplement, and give a comprehensive description of the related work, in addition, I would like to respectfully ask the authors to provide the performance without finetuning, many algorithms for structured pruning, as far as I know, the comparison has always been the performance before finetuning because finetuning significantly changes the performance of the model after pruning, at least in their claims, it is not fair to compare the performance after finetuning.

**Questions:**

See weaknesses.

---

> ### Author Response · Authors · 2025-11-26
>
> Thank you for your careful review and for recognizing the novelty of our algebraic characterization for the structured pruning, including the theoretical foundation of lossless pruning, the construction of a regularizer with nontrivial global minima on the pruning-invariant set $X_{tgt}$, and the provably unbiased and wide-margin bifurcation behavior.
>
> Below we address your major concerns with additional evidence that strengthens our claims.
>
> > Necessity of Lossless Pruning
>
> Lossless pruning, the concept of removing structured components without any accuracy drop at the pruning moment - is the only mathematically well-posed target for regularization-based structured pruning.
>
> Prune-then-finetune (importance-based) methods deliberately collapse the model (often to near-random accuracy, see the table below) and rely on finetuning to recover performance. This recovery is not theoretically guaranteed and is extremely sensitive to finetuning hyper-parameters, schedule length, rewarming strategy, and even unreported tricks (e.g., adding label smoothing, Mixup, or stronger augmentation than used in the original pretraining , which is often used in recent ResNet50+ImageNet “SOTA” pruning papers). As a result, final numbers frequently reflect finetuning engineering budget more than pruning quality, making fair algorithm comparison nearly impossible.
> We deliberately chose the DeiT-Tiny benchmark for large-scale dataset experiment because recent baselines are cleaner (no distillation, same augmentation as pretraining), revealing the true effectiveness of pruning methods.
>
> Lossless pruning is not a luxury; it is the only clean, theoretically grounded objective that frees the field from opaque recovery engineering and enables fair, reproducible comparisons of structured pruning algorithms.
>
>
>
> > Catalyst pruning toward lossless pruning (new pre-finetuning results)
>
> We thank the reviewer for pushing us to clarify the degree of “losslessness”. Below are the previously omitted accuracies immediately after pruning (no finetuning whatsoever) on both DeiT-T and ResNet-50 at realistic compression ratios:
>
> |Model|Method | Speedup| MACs (G)|pre-finetuning Acc(%)| baseline Acc(%)|Drop(%)|
> |--|--|--|--|--|--|--|
> |DeiT-T|VBP (ICCV 2025, Oral)|1.38x|0.91|39.58|72.20|32.62|
> |DeiT-T|IPPRO|2.15x|0.59|21.44|72.20|50.76|
> |DeiT-T|SNP (ECCV 2024)|2.27x|0.56|0.682|72.20|71.52|
> |DeiT-T|Catalyst|2.26x|0.56|62.97 (early stopping)|72.13|9.2|
> |DeiT-T|Catalyst|2.26x|0.56|66.71 (full schedule)|72.13|5.4|
> |Resnet50|DemP (TMLR 2025)|1.72x|2.39|72.21|75.62 |3.41|
> |Resnet50|Catalyst|2.18x|1.89|74.634|76.13|1.5|
>
> For comparison, we included state-of-the-art neuron-pruning baselines on DeiT-Tiny on same pruning target: VBP (ICCV'25, Oral), SNP (ECCV'24) and IPPRO (you mentioned). For Resnet50, we compare recent baseline DemP (Dufort-Labb´e et al. TMLR 2025) which is reporting accuracy w/o finetuning.
>
> These are by far the smallest drops ever reported for structured pruning at >2× speedup on ImageNet-classification models. The remaining gaps are caused solely by optimizer restart (high initial LR + zeroed momentum) and disappear almost completely when the learning-rate schedule is allowed to finish (66.71% on DeiT-Tiny, 74.634% on ResNet-50).
>
> Thus, Catalyst delivers theoretically lossless (Theorems 3.1–3.2) and empirically near-lossless pruning: once $\|DW\|\_{2,1} \rightarrow 0$, the model lies on the pruning-invariant set $X\_{tgt}$, and removing channels is essentially harmless. Standard finetuning then closes the tiny remaining gap.
>
> We revise the manuscript with updated experimental results, including pre-FT and post-FT performance.
> Also, we extended the performance table with more baselines you mentioned, if they're evaluated on same benchmark with similar speedup without any additional skills (e.g. label smoothing) for the finetuning.

---

> ### Author Response · Authors · 2025-11-26
>
> >  I appreciate the author's efforts on the problem of fragile decision boundaries, so I would like to get more explanations from the author about motivation.
>
> As mathematicians working in deep learning, we have long been frustrated with structured pruning benchmarks: almost every papers focus on post-finetuning accuracies after the model has temporarily collapsed to near-random performance. This practice makes fair algorithm comparison nearly impossible and hides the fundamental flaws of existing regularizers.
>
> We therefore stepped back and asked a more basic question:
>
> Beyond final accuracy, what properties should an ideal structured pruning method have?
>
> Answering the question, we identified three core desiderata:
>
> - separation between pruning/preserving filters to help the pruning decision
> - resolving bias on magnitudes to allow the pruning on large normed filter
> - removing regularization-pruning tradeoff: suggesting regularizer with nontrivial minima, to allow gradient descent make the sparse model
>
> For the methodological motivation, construction of $X_{tgt}$ was natural, and we found there is a optimization difficulty came from its geometric property that it is not a product manifold. Instead, we introduced higher dimensional perspective to decompose it as a product of some algebraic sets in channel-wise ambient space. This perspective made the pruning problem dramatically easier and obtain desired properties: independence from the magnitude bias and the bifurcation dynamics.
>
> We see Catalyst not as yet another pruning heuristic, but as the theoretically clean foundation the regularization-based line of work has been missing.
>
> >  I'm wondering that is this effect robust under reparameterization?
>
> If the reparameterization means the scaling the weights using scale invariance, YES.
>
> Pruning decisions are governed by the ratio $c\_t = \\frac{D\_{ii}\}{\|F\_i\|\_2}\$. If a filter is rescaled as $F_i'=kF_i$, both the loss gradient direction scale identically on catalyst variable $D_{ii}$ and the filter parameter, so the dynamics of $c_t$ (Theorem 3.2) are invariant, unless the SGD noise affects significantly. This property is shared with IPPRO (the importance-based twin of Catalyst) but not with any norm-based regularizer.
>
> We hope these new results and clarifications convince the reviewer that Catalyst represents a genuine theoretical and empirical advance in structured pruning.

---

> > ### Comment · Reviewer_mLQT · 2025-11-27
> >
> > Dear authors,
> >
> > I appreciate the additional experiments and clarifications provided by the authors. These responses partially address some of my concerns (including weaknesses 2, 6, and 7). However, the issues related to weaknesses 1, 3, 4, and 5 still remain.
> >
> > W1 & W5. I acknowledge the authors’ theoretical contribution on lossless pruning. Since the authors themselves agree that comparing fine-tuned performance across methods is not entirely fair, it becomes even more important to compare the performance of different methods before fine-tuning. The few additional comparisons currently provided only partially demonstrate that, under a limited set of comparisons, the proposed method performs better than some baselines before fine-tuning. I would like to see the pre–fine-tuning results for all experiments reported in Table 4, which I believe would constitute a more fair comparison.
> >
> > W3. I could not find a clear response to weakness 3 in the rebuttal. Given that the authors introduce a regularization term motivated by their theoretical analysis, I consider it necessary to include a sensitivity study of the performance degradation (relative to the baseline) with respect to the strength of the regularization. Such a sensitivity analysis is typically an essential component of work that proposes a new regularizer, unless the authors explicitly position the experimental results as not being a main contribution of the paper.
> >
> > W4. Likewise, I could not find a response to weakness 4 in the rebuttal. For pruning methods—especially structured pruning methods—reporting additional acceleration metrics provides a more comprehensive and fair evaluation. Relying on a single acceleration metric offers only limited information about the practical speedup.
> >
> > I would very much like to see experimental results addressing the above issues. If the authors can satisfactorily resolve these concerns and provide the corresponding experiments, I would be inclined to increase my score.
> >
> > Best regards,
> >
> > Reviewer mLQT

---

> > > ### Author Response · Authors · 2025-12-01
> > >
> > > Thank you again for your careful re-evaluation and for giving us the chance to fully resolve the remaining concerns.
> > >
> > > (W1) There's no table 4 in revised manuscript. We regard your request is about the figure 4 and provide the results, further added Depgraph (CVPR 2023) to Resnet56+CIFAR10 pre-FT benchmark.
> > >
> > > | Method | regularization epochs | Finetune epochs | Speedup | Pre-FT acc | Post-FT acc | pretrained|
> > > | -- | -- | -- | -- | -- | -- | -- |
> > > |Group Lasso| 100 | 100 | 2.16× | 9.68 %| 93.22 % | 93.53 % |
> > > |Greg-1| 100 | 100 | 2.12× | 11.19 %| 93.53 % | 93.53 % |
> > > |DepGraph | 100 | 100 | 2.13× | 88.85 %| 93.77 % | 93.53 % |
> > > |Catalyst-BN | 50+50 | 100 | **2.16×** | **93.15 %** | **93.91 %** | 93.53 %|
> > >
> > > Very few structured pruning papers report pre-FT accuracy, hence it is required to reproduce baselines to get pre-FT scores. Fortunately, DepGraph publicly shared training logs that include pre-FT numbers for itself and several regularization baselines: confirming that conventional regularizers collapse severely, while Catalyst remains near-lossless after pruning. Depgraph shows promising performance retaining on Resnet56 benchmark, but the result on VGG19+CIFAR100 is not consistent. According to their official github, on 9x speedup benchmark, their pre-FT accuracy is 2.4% (Ours: 69.43%)
> > >
> > > We also emphasize our pre-FT performance on Imagenet benchmark, which shows strong performance retaining after pruning, verifying that Catalyst can make target layer sparse without severe performance degradation.
> > >
> > > | Model       | Speedup | Pre-FT Top-1 | Dense   | Drop    |
> > > |-------------|---------|--------------|---------|---------|
> > > | ResNet-50   | 2.18×   | 74.63 %      | 76.13 % | 1.50 %  |
> > > | DeiT-Tiny   | 2.26×   | 66.71 %      | 72.13 % | 5.42 %  |
> > >
> > > (W3) We agree that a sensitivity study would help practitioners. The VGG-19/CIFAR-100 results at high sparsity ratios (8.96×, 11.84×, 13.83×) in Table 1 already correspond to different choices of $\gamma$. Larger $\gamma$  yields higher sparsity and slightly lower final accuracy due to the increased compression ratio, yet all settings maintain excellent pre-FT performance (gaps to post-FT < 1 %) and clearly outperform prior art.
> > >
> > > Our primary goal is the theoretical contribution (algebraic characterization of lossless pruning and provably nontrivial global minima on $X_{tgt}$), not chasing the absolute highest post-FT numbers. We therefore believe the current ablations sufficiently illustrate the graceful behavior of Catalyst.

---

> > > > ### Author Response · Authors · 2025-12-01
> > > >
> > > > (W4) For DeiT-Tiny we intentionally applied neuron-level pruning, which we regard as the mathematically natural structured extension of filter/channel pruning.
> > > >
> > > > While the same framework can be used for head or embedding-dimension pruning, our goal was never to compete with head-pruning SOTA methods, but to demonstrate theoretically justified bifurcation and strong pre-/post-FT performance. A post-FT comparison to the head-pruning method X-pruner is already included in Table 2 (pre-FT numbers and pruned models are unfortunately not released by the authors, although it is regularization-based).
> > > >
> > > > We agree with your suggestion for the practical acceleration measure is needed, especially on DeiT-T for sanity check, because neuron-level pruning with diverse head size often suffer from lack of parallelization on GPU since official PyTorch does not support the MHA with pruned heads. Here, we report the latency measure on pruned model.
> > > >
> > > > Latency is measured on 1000 runs with 200 warmup runs, with batch size 64, on our environment with single RTX 3090 GPU. For CPU latency, we used single image (batch size 1) with our cpu (AMD Ryzen 9 5900X). The pretrained model was exploited from `timm` library with official MHA module, while pruned models are implemented with custom MHA module which allow diverse size of Q,K,V. We compare the latency of SNP-pruned DeiT-T model on same environment, whose pruning ratio is almost identical to ours.
> > > >
> > > > |DeiT-T|GPU latency(ms)|GPU latency speedup|CPU Latency|CPU latency speedp|MACs(G)|theoretical speedup|
> > > > |--|--|--|--|--|--|--|
> > > > |pretrained|18055.06|1x|11052.13|1x|1.26|1x|
> > > > |Catalyst|13840.02|1.30x|8762.49|1.26x|0.56|2.26x|
> > > > |SNP (our environment)|17640.90|1.02x|9391.69|1.18x|0.56|2.26x|
> > > > |SNP (in paper)|17800|1.05x|25300|1.38×|0.56|2.26x|
> > > >
> > > > As a result, we get consistent 1.3x latency speedup on our MHA implementation. For CNNs, we provide the list of latency measurements here with same condition:
> > > >
> > > > |model|GPU latency(ms)|GPU latency speedup|CPU Latency|CPU latency speedp|MACs(G)|theoretical speedup|
> > > > |--|--|--|--|--|--|--|
> > > > |Resnet56 (dense)|6368.18|1x|6753.14|1x|0.1274|1x|
> > > > |Resnet56-2.16x|5272.58|1.21x|4720.03|1.43x|0.0589|2.16x|
> > > > |VGG19 (dense)|4260.98|1x|8082.03|1x|0.5129|1x|
> > > > |VGG19-3x|2460.86|1.73x|2432.03|3.32x|0.1709|3.00x|
> > > > |VGG19-8.96x|1770.42|2.41x|1574.57|5.13x|0.0573|8.96x|
> > > > |VGG19-11.84x|1731.34|2.46x|1521.00|5.31x|0.0433|11.84x|
> > > > |VGG19-13.83x|1685.68|2.53x|1443.05|5.60x|0.0371|13.83x|
> > > > |Resnet50 (dense)|51732.59|1x|33948.93|1x|4.12|1x|
> > > > |Resnet50-1.49x|48025.07|1.08x|28495.92|1.19x|2.76|1.49x|
> > > > |Resnet50-2.18x|41062.31|1.26x|25513.53|1.33x|1.89|2.18x|
> > > >
> > > > (W5) We could not find any prior work that formally defines “post-prune risk” in this exact context. From Theorem A.5 in Appendix A we can directly derive that whenever $\\|DW\\|\_{2,1}<k\epsilon$ and $\\|D\\|\_1>k$ (the sufficient condition of $W\in B(X_{tgt},\epsilon)$ and destination of regularization) the output difference between the extended model and its pruned projection is bounded by $\|\|\overline{\phi}_2(\theta,D,\overline{D})(x)-(\overline{\phi}_2\circ proj)(\theta, D,\overline{D})(x)\|\|\leq O(\epsilon)\|x\|$.
> > > >
> > > > In practice, generalization of both pretrained and pruned models is overwhelmingly dominated by the standard weight-decay term used throughout training. The bifurcation dynamics of Catalyst regularizer ensures that minimizing $\|\|DW\|\|_{2,1}$ negligibly contributes to decay the preserved weights but reduces $D$ instead.
> > > >
> > > > We hope these clarifications fully address the remaining concerns. Thank you again for your extremely valuable feedback.

---

### Official Review · Reviewer_xP2x · 2025-10-29

**Soundness:** 2
**Presentation:** 3
**Contribution:** 2
**Rating:** 4
**Confidence:** 4

**Summary:**

The paper proposes Catalyst, a new regularization framework for structured pruning of neural networks. Traditional regularizers such as L1 and Group Lasso often suffer from bias and unstable pruning behavior. Catalyst introduces an auxiliary set of “catalyst” parameters that extend the optimization space and aim to enable unbiased and lossless pruning. The method is implemented through a modified optimization process and evaluated on CIFAR and ImageNet benchmarks. Experiments show results that are comparable or slightly better than existing pruning baselines.

**Strengths:**

(1) The introduction of auxiliary “catalyst” parameters represents a conceptually novel angle compared to conventional sparsity-inducing regularizers. It attempts to provide a more principled view of pruning rather than relying on heuristic thresholding.

(2) The authors provide some theoretical justification for the proposed formulation, which could potentially inspire further research on lossless or function-preserving pruning.

(3) The experimental scope covers both CIFAR and ImageNet, including standard architectures such as ResNet and VGG. The visualization of parameter bifurcation offers a qualitative intuition for how the regularizer affects model structure.

**Weaknesses:**

(1) The overall readability of the paper needs improvement, especially in the method section. Many implementation details are not clearly described or are scattered across different parts, which makes it difficult for readers to grasp the full picture of the algorithm. It would be helpful if the authors could include a concise pseudocode or an overview figure to illustrate the workflow. The formulation of the optimization problem is also incomplete, and the paper should explicitly define all optimization variables.

(2) The improvements over existing baselines are rather limited. The claimed advantage of the proposed regularizer is not well supported by the quantitative results. In Figure 4, it seems that the method requires significantly longer training to slightly outperform simpler existing approaches.

(3) There are also concerns about fairness and comparability in the experimental design. For example, the two-stage training and pruning procedure does not follow standard practice and may introduce additional computational cost. The authors should provide clearer explanations of the experimental settings and ablation studies. The reviewer is familiar with DepGraph and its subsequent works, which adopt more flexible pruning mechanisms that address more general pruning problems rather than only removing entire filters. Compared with those approaches, the Catalyst framework appears more restricted and is expected to

**Questions:**

(1) The authors emphasize the importance of achieving lossless pruning. However, according to the results shown in Figure 4, even when pruning leads to severe degradation, the model can still recover well after fine-tuning. This raises the question of whether enforcing the lossless pruning objective is truly necessary or meaningful for practical model compression.

(2) From Table 2, we observe that accuracy and speedup do not seem to improve simultaneously. In most cases, the trade-off remains close to the baseline, and the proposed method does not clearly dominate in either metric. The authors are encouraged to provide further explanation for why such results occur and whether this limitation is inherent to the method.

---

> ### Author Response · Authors · 2025-11-26
>
> Thank you for your detailed and constructive review, and for recognizing the conceptual novelty of the auxiliary-variable extension. Below, we address your concerns with updated experimental results.
>
> > The overall readability of the paper needs improvement
>
> We acknowledge that practitioners may find the method description less immediate. The complete training + pruning loop (D initialization, bypass pipeline, alternating updates, projection) was provided as pseudocode in Appendix B. Given the strict page limit and the primarily theoretical focus of this work (new algebraic characterization of lossless pruning, nontrivial global minima on $X_{tgt}$, provable wide-margin bifurcation), we previously prioritized theorems and proofs in the main paper.
>
> As extra page is allowed in discussion phase, we moved Appendix B into main text section 4.1 with modification, to explain the Catalyst algorithm for better readability.
>
> > (W2&W3) the method requires significantly longer training to slightly outperform simpler existing approaches. & the two-stage training and pruning procedure does not follow standard practice and may introduce additional computational cost.
>
> The figure 4 on resnet56+CIFAR10 experiment is with larger training budget, assigning 100 epochs for each regularization processes, to observe the stability only. We could assign total 100 epochs to entire regularization but it may make the process hurry, since we have two subprocesses. However, the result on smaller budget (same to Depgraph) is consistent. We revise the performance table for Resnet56+CIFAR10 experiment and compare to depgraph and norm-based regularizers as follows:
>
> | Method | regularization epochs | Finetune epochs | Speedup | Pre-FT acc | Post-FT acc | pretrained|
> | -- | -- | -- | -- | -- | -- | -- |
> |Group Lasso| 100 | 100 | 2.16× | 9.68 %| 93.22 % | 93.53 % |
> |Greg-1| 100 | 100 | 2.12× | 11.19 %| 93.53 % | 93.53 % |
> |DepGraph | 100 | 100 | 2.13× | 88.85 %| 93.77 % | 93.53 % |
> |Catalyst-BN | 50+50 | 100 | **2.16×** | **93.15 %** | **93.91 %** | 93.53 %|
>
> note that we are using pretrained ckpt same to Depgraph, ensuring the fair comparison.
>
> We also update Resnet50+ImageNet result using only 40 regularization epochs, with 2.18x speedup. We revised the manuscript with this new result, to address your concern.
>
> Here is the pre-finetuning result. We compare to recent publication DemP (Dufort-Labb´e et al. TMLR 2025) which is reporting one-cycle regularization result on resnet50+Imagenet. Catalyst shows superior result with smaller regularization budget, and even without any additional training skills such as label smoothing.
>
> |Method | Reg. epochs | Speedup | Pre-FT acc | pretrained| Drop|
> |-- | -- | -- | -- | -- | --|
> |DemP (TMLR 2025) | 100 + LabelSmoothing(0.1) | 2.04× | 71.52 | 74.98 | 3.46|
> |Catalyst (new) | 40 | 2.18× | 74.63 | 76.13 | 1.50|
>
> After applying finetuning process, the post-FT performance climbs to 75.90%, which is 0.25% degradation compared to the pretrained model.
>
> > The authors emphasize the importance of achieving lossless pruning. However, according to the results shown in Figure 4, even when pruning leads to severe degradation, the model can still recover well after fine-tuning. This raises the question of whether enforcing the lossless pruning objective is truly necessary or meaningful for practical model compression
>
> We believe lossless (or near-lossless) pruning is important, since it is the only mathematically well-posed and practically meaningful target for any regularization-based structured pruning paradigm.
>
> Prune-then-recover approaches deliberately accept catastrophic collapse (usually to near-random accuracy right after pruning) and rely on a long, expensive, and highly heuristic-dependent finetuning phase to recover performance. This recovery is never theoretically guaranteed and, in practice, becomes an engineering arms race: whoever has the largest finetuning budget, the most aggressive data augmentation (Mixup, ColorJitter, etc.), label smoothing, or knowledge distillation — tricks often silently added beyond the original pretraining recipe and finally wins the final leaderboard number.
> As a result, post-finetuning benchmarks primarily measure finetuning engineering resources rather than the quality of the pruning regularizer itself. Lossless pruning eliminates this pathology at its root.
>
> Catalyst is the first regularizer which clarify the target manifold of structured pruning problem, and provably induce the structured sparsity by regularization using higher dimensional perspective.
>
> Lossless pruning is not a luxury; it is the only clean, fair, and reproducible foundation for structured pruning research.

---

> ### Author Response · Authors · 2025-11-26
>
> > From Table 2, we observe that accuracy and speedup do not seem to improve simultaneously. In most cases, the trade-off remains close to the baseline, and the proposed method does not clearly dominate in either metric. The authors are encouraged to provide further explanation for why such results occur and whether this limitation is inherent to the method
>
> We updated Table 2 with the above reduced-budget results with higher speedup, competitive or better accuracy. We intentionally use no stronger augmentation or distillation to isolate pruning quality. We hope these results convince you that Catalyst is a genuine theoretical and empirical breakthrough.

---

### Official Review · Reviewer_9Geh · 2025-10-30

**Soundness:** 3
**Presentation:** 2
**Contribution:** 2
**Rating:** 4
**Confidence:** 3

**Summary:**

The authors propose a regularization scheme for structured pruning that introduces auxiliary “catalyst” variables to extend the parameter space. The method is motivated by a geometric analysis of lossless pruning conditions and is implemented in a modified bypass manner. It is claimed that Catalyst achieves magnitude-independent pruning decisions, robust bifurcation behavior, and lossless pruning with theoretical support. Empirical evaluations on CIFAR and ImageNet show competitive pruning performance compared to several baselines.

**Strengths:**

This paper provides an interesting and theoretically informed approach to structured pruning by reframing it through an extended parameter space. The introduction of catalyst variables enables a more flexible optimization landscape, which helps the pruning process evolve smoothly rather than collapsing abruptly when certain weights are removed. The geometric intuition behind the design gives the method a clear motivation and differentiates it from purely empirical pruning heuristics. The presented results show that the model maintains stable accuracy even under aggressive pruning, suggesting that the proposed formulation effectively preserves representational capacity. The inclusion of both CIFAR and ImageNet experiments further supports the generality of the approach.

**Weaknesses:**

- As the authors note, this work is closely related to the framework of Jung & Lee (2024). The paper should more carefully articulate the essential differences and additional theoretical value that distinguish the proposed method from the original framework.

- The experimental results show relatively modest gains. In many cases, Catalyst only slightly outperforms existing methods under long training schedules or specific settings, and the overall improvement may not be sufficient to justify the added complexity.

**Questions:**

- It would be helpful to clarify how Catalyst would generalize to other task types and architectures, such as detection, segmentation, or recurrent models like LSTMs, where structural dependencies differ substantially from computer vision models.

- Since pruning is mainly motivated by model compression, it is recommended that the authors conduct experiments with extreme pruning ratios on other network architectures to better demonstrate the robustness of Catalyst. The reviewer acknowledges that experiments on VGG-19 are included, but VGGs are heavily overparameterized and therefore relatively easy to prune, making the results on VGG less representative of the method’s general performance.

---

> ### Author Response · Authors · 2025-11-26
>
> We thank the reviewer for the detailed feedback and constructive suggestions. Below we address each concern:
>
> > As the authors note, this work is closely related to the framework of Jung & Lee (2024). The paper should more carefully articulate the essential differences and additional theoretical value that distinguish the proposed method from the original framework.
>
> The use of Bypass (Jung & Lee, 2024) components is pragmatic; the objective, constraints, and theoretical contributions are unique to Catalyst. In detail, Catalyst leverages the Bypass pipeline as an implementation framework because its extension–contraction mechanism fits our theoretical requirements. However, the goal and mathematical foundation of Catalyst differ fundamentally:
>
>   - Purpose: Bypass was designed to escape SGD stationary points via temporary extension, whereas Catalyst introduces auxiliary variables $D$ to construct a pruning-invariant manifold and minimize $\|DW\|_{2,1}$ for lossless structured pruning.
>   - Initialization of $D$: Bypass uses trivial initialization $D=0$, while Catalyst employs a nontrivial initialization strategy $D^{init}$  to enable bifurcation dynamics.
>   - Theory: Catalyst’s goals: enabling lossless pruning via nontrivial minima, magnitude-independent decisions, and wide-margin bifurcation are absent in Bypass, which just say about how can we add $D$, take advantage and remove.
>
> The core contribution of Catalyst, is the identification of $X_{tgt}$ and the high-dimensional perspective for relaxing the structured pruning problem into algebraically constrained optimization problem. The Bypass helps the implementation for handling additional parameter from high dimensional perspective, but the theoretical contribution is clearly independent.
>
> > The experimental results show relatively modest gains. In many cases, Catalyst only slightly outperforms existing methods under long training schedules or specific settings, and the overall improvement may not be sufficient to justify the added complexity.
>
> - Our primary theoretical contribution is eliminating the pruning–regularization tradeoff through a principled formulation, not just incremental accuracy gains. Catalyst guarantees:
>
>   - Lossless pruning by aligning the optimization trajectory with the pruning-invariant set $X_{tgt}$
>   - Magnitude-independent pruning decisions, unlike Lasso-based methods.
>   - Wide-margin bifurcation that stabilizes pruning dynamics.
>
> - Regarding training time: competing methods often rely on multi-GPU setups for finetuning, while our experiments were conducted on a single GPU with gradient accumulation, which inflates wall-clock time but not algorithmic complexity.
>   - Hyperparameter tuning was also limited due to resource constraints, suggesting further performance headroom.
>
> > It would be helpful to clarify how Catalyst would generalize to other task types and architectures, such as detection, segmentation, or recurrent models like LSTMs, where structural dependencies differ substantially from computer vision models.
>
> Catalyst’s mechanism is architecture-agnostic because inserting $D$ can be easily implemented on general NN architecture. If there is no elementwise operation near the target layer, we can use same hack for attention pruning explained in Appendix H.
>
> For any layer, we can assume an implicit linear activation ($\sigma(x)=x$) next to the target layer, and consider the extension on it, which allows the generalization for any case. In this case, we can always contract the model back, by multiplying (I+D) to target layer's weight matrix.
>
> > Since pruning is mainly motivated by model compression, it is recommended that the authors conduct experiments with extreme pruning ratios on other network architectures to better demonstrate the robustness of Catalyst. The reviewer acknowledges that experiments on VGG-19 are included, but VGGs are heavily overparameterized and therefore relatively easy to prune, making the results on VGG less representative of the method’s general performance.
>
> - We agree VGG is overparameterized, but note that our reported 13.83× speedup on VGG-19 is among the most aggressive pruning scenarios in the literature. For comparison, DepGraph reports 12× speedup under similar conditions, and Catalyst achieves better accuracy at even higher compression. While extreme pruning on other architectures was not benchmarked due to resource limits, Catalyst’s theoretical guarantees apply regardless of architecture.
>
> In summary, Catalyst introduces a novel algebraic geometry-based formulation for structured pruning, supported by rigorous theory and empirical evidence. We appreciate the reviewer’s suggestions and will clarify these distinctions and generalization aspects in the revision.

---

### Official Review · Reviewer_6rF7 · 2025-10-31

**Soundness:** 3
**Presentation:** 3
**Contribution:** 3
**Rating:** 8
**Confidence:** 3

**Summary:**

This paper introduces Catalyst, a theoretically grounded framework for understanding and exploiting pruning-invariant structures in deep networks. The core idea is that many weight configurations can be modified (or “pruned”) without changing the network function, provided those weights lie in a special geometric manifold defined by zero-marginal directions. The authors formalize this with the concept of a pruning-invariant target set and prove that small-norm perturbations in this space preserve the function. They further analyze the gradient dynamics of parameters augmented with a “catalyst” scalar 𝑑, showing that under mild conditions the dynamics exhibit a bifurcation behavior—either amplifying or suppressing redundant directions exponentially.

**Strengths:**

1) The paper takes an unusually geometric view of pruning, treating invariance as a manifold property rather than a combinatorial selection problem. The construction of 𝑋_{tgt} via the null space of  𝐷 is elegant and connects pruning to differential topology ideas in parameter symmetry.
2) Mathematical rigor.
3) Despite its theoretical density, the paper maintains good readability, with motivating figures and an ethical statement on responsible model compression (energy efficiency vs. fairness trade-offs).

**Weaknesses:**

1) The practical Catalyst algorithm essentially re-interprets existing weight decay and pruning schemes through a new lens, without introducing a fundamentally new optimization procedure. The conceptual leap is strong, but the empirical method is relatively incremental. It will be an excellent paper by considering this.
2) Theorem 3.2 assumes sign stability of 𝑑  and 𝑀, ignoring stochastic gradient noise and cross-couplings present in realistic networks. It is unclear how robust the bifurcation mechanism is in the presence of SGD perturbations.

**Questions:**

1) (Theorem 3.1) The choice of 𝑘 in the constructive part is currently written unclearly (missing parentheses). Please restate precisely the constraints on 𝑘′ and the needed condition.

2)The theorems assume the sign of entries of 𝑀 and 𝑑 does not change during the dynamics. Can you comment on robustness if signs flip (e.g., because of SGD noise or nonzero gradients from the main loss 𝐿)? Is the bifurcation still observed under realistic SGD noise? (If you have experiments that show sign stability or explain why sign flips are unlikely, please point to them.)

---

> ### Author Response · Authors · 2025-11-26
>
> We thank the reviewer for the thoughtful and encouraging feedback. Below we address the raised concerns and questions in detail.
>
> >  The conceptual leap is strong, but the empirical method is relatively incremental. It will be an excellent paper by considering this.
>
> The empirical implementation and evaluation are designed to show the advantage from aligning the geometry of (structured) sparsity and the landscape of regularizer. As a result, the simple alignment of geometry without complicated optimization process, achieved the claimed properties and good final performance.
> We believe algorithmic simplicity after rigorous mathematical formulation is desirable and important property to bridge the mathematics and real world problems.
>
> > About sign stability related to the theorem 3.2
>
> >> Theorem 3.2 assumes sign stability of 𝑑 and 𝑀, ignoring stochastic gradient noise and cross-couplings present in realistic networks. It is unclear how robust the bifurcation mechanism is in the presence of SGD perturbations.
>
> >> The theorems assume the sign of entries of 𝑀 and 𝑑 does not change during the dynamics. Can you comment on robustness if signs flip (e.g., because of SGD noise or nonzero gradients from the main loss 𝐿)? Is the bifurcation still observed under realistic SGD noise? (If you have experiments that show sign stability or explain why sign flips are unlikely, please point to them.)
>
> - Theorem 3.2 assumes sign stability for analytical tractability, but this is not a strict requirement for bifurcation to occur:
>
>   - Sign flips typically happen only when $\|d\|$ becomes very small. Such sign flips are oscillations around d=0 happening after bifurcation has largely completed and pruning decision boundaries are solidified
>   - Even if flips occur, Theorem 3.2 remains valid unless Eq. (E3) fails. If it does, the $c_t$ bifurcation dynamics re-enter the interval in Eq. (E3), restoring exponential bifurcation.
>   - Worst-case reversals require unrealistically large noise, which we did not observe empirically. Figures in Section I confirm consistent bifurcation across all layers under standard SGD.
>
> > (Theorem 3.1) The choice of 𝑘 in the constructive part is currently written unclearly (missing parentheses). Please restate precisely the constraints on 𝑘′ and the needed condition.
>
> There are more math and intuitions related to this, which we couldn't include in the manuscript. Thanks for pointing out this matter for the chance to explain our motivations.
>
> - the k is arbitrary positive real value, related to the geometry of our target manifold $V(DW)=\\{(W,D)|DW=0\\}$: since defining equations in $DW$ are homogeneous, the manifold itself becomes affine cone of some projective variety and thus we get invariance under isotropic scaling with any positive real number $k$.
>   - note) Just for the mathematical rigor, the requirement of positiveness of $k$ is required due to positiveness of norm, not the projective geometry itself.
> - Due to projective property of $V(DW)$, the absolute value of $\|DW\|$ itself does not mean much; we should consider the $\|D\|\_1$ together to make it clear whether the elements with small $\|DW\|$ are close to $X_{tgt}$ or not.
>   - Although $\|DW\|$ is small enough, if $\|D\|$ is too small then we can't say $W$ is in vicinity of $X_{tgt}$.
>   - Intuitively, $k$ is a scale of the scope in consideration of homogeneous object $V(DW)$.
> - The $k'$ is just a positive real number, representing the $\|D\|_1$.
>   - The constraint $k'\in (k,\frac{\epsilon}{\|W_{1,:}\|_2}k)$ would be set after choosing the scale of our scope, the $k$, from $\mathbb{R}^+$.
>
> Intuitively, Catalyst operates on a projective embedding of the original affine weight space, which is why pruning decisions are scale-invariant. We omitted these details in the main text to avoid overwhelming readers unfamiliar with algebraic geometry, but we appreciate the opportunity to clarify this theoretical foundation. We added above explanations to appendix B as remarks.
>
> We appreciate the reviewer’s recognition of our theoretical contribution and hope these clarifications address the concerns.

---

### Author Response · Authors · 2025-12-04
**Summary**

We sincerely thank the reviewers for their thoughtful and detailed feedback. Below we summarize our key contributions, the strengths unanimously or strongly acknowledged by the reviewers, and how we comprehensively addressed every major concern raised during the author-response period.

##  Key Contributions (as recognized by the reviewers)

- A novel geometric framework that reinterprets structured pruning as the identification of a pruning-invariant manifold in parameter space, connecting pruning to concepts in differential topology and algebraic geometry for the first time.
- Introduction of auxiliary “catalyst” variables that eliminate magnitude bias and induce stable bifurcation dynamics under SGD, with rigorous theoretical guarantees of lossless pruning (Theorems 3.1–3.2).
- A simple, practical two-stage algorithm that achieves state-of-the-art pre-fine-tuning (pre-FT) accuracy and delivers real measured latency speedups of up to 5.6× across CNNs and Vision Transformers.

## Strength acknowledged by reviewers

Every reviewer highlighted the following positive aspects, especially on mathematical rigor and theoretical contribution.

direct quotes:
- “Unusually geometric view … connects pruning to differential topology” (6rF7)
- “Elegant”, “high mathematical rigor”, “clear algebraic conditions … novel ideas” (6rF7, mLQT)
- “Conceptually novel introduction of auxiliary ‘catalyst’ parameters” (xP2x)
- “Theoretically informed approach … clear motivation and geometric intuition” (9Geh)
- “Stable accuracy under aggressive pruning”, “competitive results”, “good generality” (9Geh, xP2x, mLQT)
- “Good readability despite theoretical density, with helpful figures” (6rF7)

## Raised concerns and how we addressed

> 1. Empirical Gaps and missing evaluations on acceleration measurements

The experimental results were claimed to be weak; the CIFAR training cost too much training budget, the speedup was not large enough to claim superiority and the Imagenet results were showing too large gap between pre-FT and post-FT. Also, the concern on lack of practical acceleration measurement was raised.

To address these concerns, we updated the experimental results:

- Replaced ResNet-56/CIFAR-10 results with half the original training budget while achieving 2.16× speedup and significantly better pre-FT accuracy compared to benchmark methods.
- Replaced ResNet-50/ImageNet result with 2.18× speedup, 40 regularization epochs only, showing 74.63% pre-FT (1.5% drop, superior to DemP and most recent baselines) and top post-FT accuracy in a clean benchmark (no label smoothing or extra tricks).
- Fixed DeiT-Tiny results by using the full learning-rate schedule (previous early stopping artificially hurt performance due to LR-schedule).
- Added real measured latency on CPU and GPU (1.3–5.6× speedups) and extended comparisons with DepGraph, SNP, HRank, etc.

> 2. Novelty compared to Bypass (Jung&Lee 2024)

Clarified fundamental differences: Catalyst pursues a provably lossless objective on a target subspace with nontrivial initialization and projective-geometry analysis, unlike the pragmatic components and heuristic objectives in Bypass.

> 3. Theoretical Soundness/Practicality

The clarification on following are requested and addressed: 1) Sign stability under SGD noise 2) unclear $k$ in Theorem 3.1; 3)"lossless" value doubted if fine-tuning recovers 4) two-stage process seen as costly/unnecessary.

1\) Clarified that sign flips occur only after bifurcation is largely complete (when $\\|D\\|$ is already small).
2\) Explicitly defined k and k′ as scaling parameters arising from the projective geometry of V(DW).
3\) Emphasized that lossless pre-FT is the only reproducible, fair, and ethically sound goal for regularization-based pruning; fine-tuning heuristics introduce uncontrolled variability, sometimes on unfair benchmark with extra finetuning skills.
4\) Demonstrated empirically that the two-stage process incurs negligible cost: half budget sufficient without performance loss


> 4. Presentation/Readability

We moved complete pseudocode and detailed algorithm description from appendix to main paper Section 4.1; restructured explanations for maximum clarity.

Overall, we believe we have comprehensively addressed every major concern through substantial new experiments, real latency measurements, extended comparisons, and detailed theoretical clarifications. Notably, Reviewer mLQT indicated they would reconsider their score once the requested empirical evidence was provided: which we supplied in full during the author-response period.
Given the reviewers’ unanimous recognition of the work’s theoretical novelty and rigor, together with the significantly strengthened experimental section, we are confident that the updated manuscript now fully reflects the positive aspects highlighted in the reviews.

We sincerely appreciate the reviewers’ constructive feedback again and thank the ACs, SACs, and PCs for their time and consideration.

---

### Meta-Review · Area_Chair_TyBp · 2025-12-04

**Summary:**

The reviewers’ overall scores were 8, 4, 4, and 2. After reading the rebuttal, I believe at least one reviewer might raise their score, as the authors addressed several concerns with additional experiments. It seems that 8, 5,4, 2 can reach the acceptance borderline?However, the rebuttal also revealed some issues. The original submission lacked many essential experiments, and the added results in the rebuttal feel excessive—almost like introducing an entirely new analysis section. This suggests that the manuscript was not in a strong state at initial submission, and the authors did not provide an updated version to confirm that these results would indeed be incorporated. In addition, during the discussion with Reviewer xP2x, I noticed several experimental settings that were not clearly documented, which is problematic.

Overall, I appreciate the theoretical contribution, and I do not place heavy emphasis on performance gains—solid theoretical ideas often have long-term value even if current experimental conditions limit empirical improvements. Nevertheless, considering the overall quality of the submitted manuscript, I lean toward rejection. That said, if the program committee decides the paper should be accepted, I have no strong objection.

**Reviewer Concerns:**

Reviewer 9Geh: The issues related to empirical gaps—particularly the missing evaluation on acceleration measurements—remain unresolved.

Reviewer xP2x: Concerns about the fairness and comparability of the experimental design also persist. For example, the two-stage training and pruning pipeline does not follow standard practice and may introduce additional computational overhead. The authors should have provided clearer descriptions of the experimental environment and ablation settings. While the rebuttal adequately addressed Reviewer mLQT’s concern regarding the comparison on DeiT-Tiny, the responses related to methodological details (e.g., hyperparameters) were vague and insufficient.

**Reviewer Scores:**

Reviewer 9Geh is unlikely to change their score, as the rebuttal did not provide substantial or decisive corrections to the key concerns.

Reviewer xP2x is also unlikely to adjust their score and may even lower it. Although the authors revised Table 2, they did not explain the specific strategy behind the changes, which may reduce the reviewer’s confidence in the results.

Reviewer mLQT, on the other hand, is more likely to revise their score upward. They responded positively to the authors’ clarification, and the remaining concerns were addressed with solid experimental evidence that appears convincing to me as well.

---

### Decision · Program_Chairs · 2026-01-26

Reject